

# Tidal and seasonal forcing of dissolved nutrient fluxes in reef communities

Renee K. Gruber[1], Ryan J. Lowe[2,3], and James L. Falter[2,3]

[1]The Australian Institute of Marine Science, Townsville, Queensland, 4810, Australia
[2]The Oceans Institute, University of Western Australia, Crawley, Western Australia, 6009, Australia
[3]The ARC Centre of Excellence for Coral Reef Studies, Crawley, Western Australia, 6009, Australia

*Correspondence to*: Renee K. Gruber (r.gruber@aims.gov.au)

**Abstract.** Benthic fluxes of dissolved nutrients in reef communities are controlled by oceanographic forcing including hydrodynamic regime and seasonal changes in oceanic nutrient supply. Up to a third of reefs worldwide can be characterised
as having circulation that is predominantly tidally-forced, yet almost all previous research on reef nutrient fluxes has focused on systems with wave-driven circulation. Fluxes of dissolved nitrogen and phosphorus were measured on a strongly tide-dominated reef platform with a spring tidal range exceeding 8 m. Nutrient fluxes were estimated using a one-dimensional control volume approach, combining flow measurements with modified Eulerian sampling of waters traversing the reef. Measured fluxes were compared to theoretical mass-transfer-limited uptake rates derived from flow speeds. Reef communities
released a moderate amount of nitrate, potentially derived from the remineralization of phytoplankton and dissolved organic nitrogen. Nutrient concentrations and flow speeds varied between the major benthic communities (coral reef and seagrass), resulting in spatial variability in estimated nitrate uptake rates. Rapid changes in flow speed and water depth are key characteristics of tide-dominated reefs, which caused mass-transfer-limited nutrient uptake rates to vary by an order of magnitude on time scales of ~minutes – hours. Seasonal nutrient supply was also a strong control on reef mass-transfer-limited
uptake rates, and increases in offshore dissolved inorganic nitrogen concentrations during the wet season caused an estimated twofold increase in uptake.

## 1 Introduction

Reef organisms remove nutrients from overlying waters for essential metabolic and biogeochemical processes, which enable them to accumulate biomass and ultimately support broader marine food webs (McMahon et al., 2016;Parrish, 1989). Reef
waters have carbon concentrations that are orders of magnitude greater than nitrogen (N) and phosphorus (P), and thus benthic community productivity is generally limited by the rates at which organisms can acquire N and P (Atkinson and Falter, 2003;Larned, 1998;Smith, 1984). Suspended N and P can be categorized in dissolved inorganic (DIN, DIP), dissolved organic (DON, DOP), and particulate organic (PON, PP) fractions, which are generally utilized by different groups of organisms. Primary producers take up labile (readily reduced and incorporated into new tissue) dissolved inorganic nutrients in the forms
of nitrate/nitrite (NOx), ammonium (NH4+), and phosphate (DIP), which are found at low concentrations in reef waters.





Ocean-derived dissolved organic N and P compounds are generally thought to be refractory or energetically-intensive for organisms to utilise (Knapp et al., 2005); thus, DON tends to dominate the nitrogen pool and DOP concentrations are generally low and similar to DIP (Furnas et al., 2011). However, studies on DON uptake have provided mixed results: some have measured a net production of DON by reef communities (Cuet et al., 2011a;Tanaka et al., 2011), while others have found

evidence that primary producers (Vonk et al., 2008) and filter-feeders (Rix et al., 2017) can directly utilize some DON compounds. Finally, particulate N and P pools in reef waters are generally dominated by small phytoplankton ($< 2\mu$m) and bacterial cells, and are an important source of nutrients for reef suspension and filter-feeding organisms (Houlbrèque et al., 2006;Ribes et al., 2005;Wyatt et al., 2010).

The majority of studies on reef nutrient dynamics have focused on the labile dissolved inorganic species as these are tightly coupled to reef productivity (D'Elia and Wiebe, 1990;Szmant, 2002). Research over the last two decades has shown that the upper limit of DIN and DIP uptake on reefs is physically constrained by mass-transfer, a term that refers to the transfer of solutes in the water column across diffusive boundary layers surrounding the tissue surface of an organism (Bilger and Atkinson, 1992;Hurd, 2000). Nutrient uptake in reef waters is typically mass-transfer limited (i.e. the biological demand for

nutrients is higher than the physical rate at which they can be supplied). Therefore, the uptake rate has a first-order relationship with nutrient concentration and is a function of water velocity, bottom roughness properties, and diffusion characteristics of the solute (Atkinson, 2011). Due to dependency of mass-transfer-limited nutrient uptake on flow speed, the local hydrodynamic conditions within a reef directly affect uptake rates of DIN and DIP (Atkinson and Bilger, 1992;Baird et al., 2004;Falter et al., 2016;Reidenbach et al., 2006;Thomas and Atkinson, 1997), and these uptake rates can be predicted for a

particular reef given sufficient information (Falter et al., 2004;Zhang et al., 2011). However, validating these models with observations from living systems remains a major challenge as measurements must occur at spatial and temporal scales relevant to reef circulation, and in situ uptake is often confounded by simultaneously-occurring biogeochemical processes that release DIN and DIP to the water column (Atkinson and Falter, 2003;Wyatt et al., 2012).

Accurate measurements of nutrient uptake in natural reef communities are still relatively limited, and are just beginning to incorporate spatial and temporal variability in forcing conditions (Lowe and Falter, 2015), such as gradients in wave energy across a reef or seasonal changes in local oceanic nutrient concentrations (e.g., Wyatt et al., 2012). While many studies have assessed nutrient dynamics in reefs experiencing long-term nutrient enrichment (Cuet et al., 2011a;Furnas, 2003;Paytan et al., 2006;Tait et al., 2014), relatively little work has focused on systems experiencing natural pulses in nutrient delivery from

processes such as coastal upwelling (Andrews and Gentien, 1982;Stuhldreier et al., 2015;Wyatt et al., 2012) or internal waves (Green et al., in press;Leichter et al., 2003;Wang et al., 2007). Additionally, the majority of reef research to date has occurred on reefs whose circulation patterns and residence times are mainly driven by wave-breaking on the forereef (Monismith, 2007). However, the circulation of up to a third of reefs worldwide has been estimated to be tide-dominated, defined as the case where annual mean significant wave height (offshore of the reef) is less than the mean tidal range (Lowe and Falter, 2015). Reefs



that are strongly tide-dominated can experience substantial variability in flow speeds and water depths over a single semidiurnal tidal cycle (Lowe et al., 2015), which suggests that mass-transfer-limited nutrient uptake rates (and other biological processes) would also vary throughout the tidal cycle.

The Kimberley coastal region (located in remote northwest Australia) has a macrotidal regime where spring tidal ranges can reach 12m in some locations (Kowalik, 2004). The region contains thousands of islands with a total reef area estimated to be ~2000 km$^2$ (Kordi and O'Leary, 2016) inhabited by diverse coral reef and seagrass communities (Richards et al., 2015;Wells et al., 1995). Recent work has revealed the strongly tide-dominated circulation that can occur on Kimberley reef platforms (Lowe et al., 2015). When the tidal amplitude (half the tidal range) is greater than the reef elevation relative to mean sea level,

water levels drop below the reef for portions of each tidal cycle, and this "truncation" of the semi-diurnal tide results in asymmetric phase durations (~10 hour ebb and ~2 hour flood) and flow speeds (Lowe et al., 2015). Extended periods of low water depth on reef platforms such as Tallon Island can cause communities to experience high irradiances that result in diel temperature changes up to 11° C (Lowe et al., 2016) and dissolved oxygen fluctuations among the most extreme measured worldwide (Gruber et al., 2017). Recent measurements of coral calcification (Dandan et al., 2015), seagrass productivity

(Pedersen et al., 2016), reef community metabolism (Gruber et al., 2017), and particulate nutrient uptake (Gruber et al., 2018) have been published from tide-dominated systems, yet little is currently known about how these large tides control fluxes of dissolved nutrients. The objectives of this study were to: 1) measure fluxes of dissolved N and P on a tidally-forced reef, 2) compare measured rates to maximum potential uptake predicted by mass-transfer theory, and 3) compare tidal forcing (velocity and water depth changes) and oceanic forcing (seasonal changes in nutrient concentration) of mass-transfer-limited uptake

rates. This work will provide some preliminary insight into the magnitudes, variability, and temporal scales of nutrient cycling on tide-dominated reefs.

## 2 Methods

### 2.1 Field site

A series of field experiments were conducted in the western Kimberley region at Tallon Island, which contains a large intertidal

reef platform (surface area 2.2 x 10$^6$ m$^2$) on its eastern side (Figure 1). The platform is elevated slightly (25 cm) above mean sea level, and the seaward rim is 10 cm shallower than the rest of the platform; this feature, coupled with bottom friction, prevents reef benthic communities from becoming emersed during low tide (Lowe et al., 2015). The platform is covered with a series of regular shore-parallel ridges ~0.15 - 0.25 m in height and contains two benthic communities: a seagrass-dominated inner zone (from the fringing mangrove shoreline to 400 m landward of the reef crest), and a coral reef outer zone (200 m wide

extending shoreward from the crest). Between these distinct communities, a 200 m zone of rubble and sand occurs where the seagrass and coral reef communities mix (Figure 1). *Enhalus acoroides* is found with *Thalassia hemprichii* in the seagrass



zone (Wells et al., 1995). The coral community contains brown foliose macroalgae (predominantly *Sargassum* spp.), a diverse assemblage of small hard corals (~5-10% cover), soft coral, coralline macroalgae, and crustose coralline algae.

The Kimberley region experiences a sub-tropical climate, so field experiments at Tallon reef were conducted during the dry
(5 – 20 October 2013) and wet seasons (4 – 9 February 2014). Nutrient concentrations were measured from duplicate filtered water samples (Table 1) and were collected around hydrodynamic instrumentation, forming a one-dimensional control volume as detailed below (see also Gruber et al., 2017). This approached allowed estimation of dissolved nutrients fluxes across the reef benthos, which represent the net uptake or release of nutrients. Estimates of uptake of DIN and DIP at the limits of mass-transfer were made using hydrodynamic data over a spring-neap cycle (~15 days) collected during the hydrodynamic study of
Lowe et al. (2015) and nutrient concentrations from water sampling during the Oct and Feb field experiments. This manuscript presents tidal phase-averaged data as a way to visualize measurements that tend to fluctuate with the phase of tide. Phase-averaged values in this study are ensemble averages from every point in the semidiurnal (M2) tidal cycle (e.g., average of all measurements taken during low tide).

## 2.2 Dissolved nutrient sampling

Water samples were collected during both field experiments for analysis of dissolved nutrient concentrations in offshore and reef flat waters. Eulerian sampling occurred at three stations (Figure 1): the coral zone ('CR'), the seagrass zone ('SG'), and offshore of the reef in adjacent waters ('Off'). Offshore samples were collected from just beneath the water surface throughout the semidiurnal tidal cycle on days of sampling (Table 1). Collecting water samples on the reef platform was not feasible during periods of peak flood and ebb, which occurred 0 – 1 and 4 – 6 hours, respectively, after the onset of reef flooding (when
offshore waters first overtopped the reef crest). Rapid changes in water depth during these tidal phases caused current speeds exceeding 0.8 m s$^{-1}$ (Figure 2), which made for unsafe conditions for sampling by foot or boat. Reef sampling was conducted during the remaining 9 hours of each tidal cycle, either by foot when water depths were low (~0.4 – 0.6 m) or by boat during high tide (1 – 4 hours from the onset of reef flooding).

Water samples were collected for analysis of dissolved nutrients with a 50 mL syringe (pre-rinsed with reef water) and immediately filtered (Minisart, pore size 0.45 $\mu$m) into 30 mL pre-rinsed tubes. These samples were placed in darkness on ice and were frozen upon return to the field station (several hours); samples were transported and stored frozen until analysis at the laboratory (<4 weeks from the end of the field experiment). Analyses of nitrate and nitrite (NO$_x$), ammonium (NH$_4^+$), and inorganic phosphorus (DIP) concentrations were determined on a flow-injection autoanalyzer (Lachat QuikChem 2500) using
standard methods (Strickland and Parsons, 1972). Total dissolved nitrogen was determined by persulfate oxidation of filtered samples (Valderrama, 1981) followed by analysis of nitrate as above. Dissolved organic nitrogen (DON) was estimated from the total dissolved nitrogen less NO$_x$ and NH$_4^+$. All nutrient concentrations presented are the mean of duplicate samples.



## 2.3 Control volume approach

The control volume (CoVo) technique utilises flow measurements and modified Eulerian sampling of solutes or particles to derive in situ benthic flux estimates. Tallon reef platform is well-suited to a one-dimensional CoVo approach due to long periods (approximately 10 h of each semidiurnal tidal cycle) of consistent flow direction; nutrient sampling may thus be

conducted at 'upstream' and 'downstream' sites during these periods. A similar approach has previously been used on Tallon reef to estimate its benthic metabolism (Gruber et al., 2017) and particulate material uptake (Gruber et al., 2018) rates. A bottom-mounted acoustic Doppler current profiler (Nortek Aquadopp HR) was stationed near SG (Figure 1) and measured current velocity and water depth ($h$) at 1 Hz and 0.03 m bins. Depth-averaged flow speeds ($u$) were bin-averaged at 5 min intervals. During the reef's extended ~10 hour ebb tide, water drained off the platform in a consistent northeast direction (80°

$\pm$ 30°, mean $\pm$ standard deviation), along which the water sampling stations were aligned. Depth-averaged current velocity was rotated in this ebb flow direction ($u_x$) and transport $q_x$ was estimated as

$$q_x = u_x h, \tag{1}$$

assuming negligible horizontal dispersion. The net flux $J_{net}$ (in mmol N or P m$^{-2}$ d$^{-1}$) of each nutrient species (NO$_x$, NH$_4^+$, DIP, and DON) into the benthos was estimated as

$$-J_{net} = \bar{h}\frac{d\bar{C}}{dt} + q_x\frac{(C_{CR} - C_{SG})}{dx}, \tag{2}$$

where the distance between sampling stations $dx$ was 540 m and $\bar{h}$ was the mean water depth along $dx$. Nutrient concentrations at CR and SG are represented by $C_{CR}$ and $C_{SG}$, respectively; $\bar{C}$ is the mean of concentrations at both stations (Genin et al., 2002). Positive values of $J_{net}$ represent net benthic nutrient uptake and negative $J_{net}$ indicates net release of nutrients to the water column. The 'local' benthic flux (i.e., nutrient uptake or release occurring at a sampling station) is represented by the

first right-side term of Eq. (2), and was estimated at hourly intervals when water sampling occurred. The second term of Eq. (2) represents the 'advective' flux (i.e., nutrient uptake or release during transit between sampling stations), and was averaged over a time interval that fluctuated with changing flow speeds. These estimates were then linearly interpolated to times where local estimates existed; this method is described in greater detail in Gruber et al. (2017).

## 2.4 Uptake rates at the limits of mass-transfer

For comparison with the field observations, the theoretical uptake rates of DIN and DIP at the limits of mass-transfer ($J_{MTL}$) were calculated for each of the measurements of $J_{net}$ above. Assuming nutrient concentrations at the tissue surface of benthic organisms were near zero, $J_{MTL}$ was estimated along the study transect (from SG to CR) as (Falter et al., 2004)

$$J_{MTL} = S\bar{C}, \tag{3}$$

where $S$ is the mass-transfer velocity (in m d$^{-1}$). Estimates of $J_{MTL}$ and $S$ were made for NO$_x$, NH$_4^+$, and DIP, and were averaged

over the same time intervals as $J_{net}$. Mass-transfer velocity $S$ was estimated as (Falter et al., 2004)

$$S = uC_D^{0.5}/(Re_k^{0.2}Sc^{0.6}), \tag{4}$$



where $C_D$ is the drag coefficient, $\mathrm{Re_k}$ is the roughness Reynolds number, and Sc is the Schmidt number. The Schmidt number is defined as the kinematic viscosity ($v$) divided by the diffusivity of the nutrient species (Li and Gregory, 1974). The drag coefficient $C_D$ increases dramatically as reef water depth decreases (Lentz et al., 2017), and so was estimated from empirical relationships using $h$ and height of reef ridges (McDonald et al., 2006), following the same approach as used in estimates of

reef metabolism (Gruber et al., 2017). The roughness Reynolds number $\mathrm{Re_k}$ is defined as

$$\mathrm{Re_k} = u_* k_s / v, \qquad (5)$$

where $k_s$, a hydraulic roughness length scale, was 0.5 m (Lowe et al., 2015) and the shear velocity $u_*$ is a function of bottom shear stress $\tau_b$ and seawater density $\rho$ as

$$u_* \equiv \sqrt{\tau_b/\rho} = u\sqrt{C_D/2}. \qquad (6)$$

Estimates of maximum potential nutrient release ($J_{\mathrm{release}}$) represent the flux of $NO_x$, $NH_4^+$, and DIP necessary to match the observed $J_{\mathrm{net}}$ assuming uptake occurred at mass-transfer-limited rates, and were estimated as (Wyatt et al., 2012)

$$J_{\mathrm{release}} = J_{\mathrm{net}} - J_{\mathrm{MTL}} \qquad (7)$$

for each of the intervals over which $J_{\mathrm{net}}$ was calculated.

Large changes in water depth, flow speed, and nutrient concentration occurred during each tidal cycle, yet measurements of $J_{\mathrm{net}}$ could only be made during ebb tide (generally 6 – 12 hours after onset of reef flooding). In order to understand how the range of flow speeds experienced by this reef platform could influence maximum potential nutrient uptake rates, we calculated $J_{\mathrm{MTL}}$ continuously over a full ~15 day spring-neap cycle at SG and CR. Current speed was not measured at CR during the Oct 2013 field experiment, so measurements from a detailed hydrodynamic study (which included instruments positioned at SG

and CR) during Apr 2014 were used (Lowe et al., 2015). Flows on the reef platform are strongly tide-driven, and can be predicted based on water depth and tidal phase; given that spring and neap tidal ranges were very similar between Oct and Apr experiments, velocity measurements from Apr can be considered representative of velocities in Oct. Tidal phase-averaged concentrations of $NO_x$, $NH_4^+$, and DIP were approximated for both sites (CR and SG) and field experiments (Oct and Feb) using measured concentrations (Figure 3) where available. As it was not possible to collect water samples during peak ebb

tide (due to hazardous conditions), nutrient concentrations in offshore waters (Table 2) were assumed to be representative of concentrations on the reef platform during those times. Tidal cycle averages of mass-transfer velocities ($S_{\mathrm{cyc}}$) and mass-transfer-limited nutrient flux ($J_{\mathrm{cyc}}$) were calculated as the mean of all $S$ and $J_{\mathrm{MTL}}$, respectively, occurring within an individual semidiurnal tidal cycle beginning when water flooded the reef platform.

Uncertainties in estimates of $S$, $J_{\mathrm{net}}$, and $J_{\mathrm{MTL}}$ were estimated by propagating standard deviations using Monte Carlo simulation

(10,000 iterations). Error terms for hydrodynamic variables were derived from bin-averaged data (Lehrter and Cebrian, 2010).



## 3 Results

### 3.1 Nutrient concentrations and measured fluxes

Characteristics of offshore water (temperature, salinity and nutrient concentrations) showed some differences between dry and wet season field experiments. Water temperature was ~3° C warmer during the wet season in Feb, and levels of DIN were

elevated, with $NO_x$ concentrations approximately double those measured during the dry season in Oct (Table 2). Salinity and concentrations of DIP and DON were slightly lower during the wet season in Feb. Reef platform nutrient concentrations were similar to offshore concentrations during flood tide and the start of ebb tide (~3 – 6 hours after reef flooding, Figure 3); during the remaining 6 hours of ebb tide, the concentrations of DIN changed dramatically depending on the reef zone (benthic community type). In the case of $NO_x$, concentrations decreased in the seagrass zone (SG) but increased in the coral zone (CR)

by up to five times compared to offshore levels (Figure 3a,b). Increases in $NH_4^+$ occurred at both SG and CR during ebb tide (Figure 3c,d), while DIP was generally lower than offshore concentrations but tended to increase at CR during the final few hours of ebb tide (Figure 3e,f).

Fluxes of DIN and DIP estimated using the CoVo technique were generally negative, indicating a net efflux (release) of

nutrients from the benthos to the water column. This was especially true for $NO_x$, where net nutrient release ($J_{net} < 0$) reached 5 mmol m$^{-2}$ d$^{-1}$ (Figure 4a), and net uptake ($J_{net} > 0$) was not observed during any point in either field experiment. Fluxes of $NH_4^+$ and DIP varied between net uptake and release (Figure 4b,c), and $J_{net}$ for DIP tended to transition from net uptake to net release over the duration of ebb tide. There were no substantial differences in overall mean $J_{net}$ of dissolved inorganic nutrients between Oct and Feb field experiments (Table 3). Fluxes of DON did differ between seasons; $J_{net}$ varied between net uptake

and net release during Oct (Figure 4d) although mean $J_{net}$ was negligible (Table 3). During Feb, $J_{net}$ of DON transitioned from net uptake to net release over the ebb tide (Figure 4d), but showed a large uptake on average (Table 3).

### 3.2 Mass-transfer velocity and nutrient uptake

The mass-transfer velocity $S$ is a function of flow speed and is indirectly related to water depth through the drag coefficient; the magnitude of $S$ depends on the diffusivity of the nutrient species of interest (though the Schmidt number) yet is unrelated

to nutrient concentration (Eq. 4). For simplicity, only values of $S$ for $NO_x$ are shown, as the values of other species ($NH_4^+$, DIP) differ only in magnitude by a constant factor (due to diffusivity). Although temperature influences $S$ through viscosity, changes in temperature on the reef platform had a negligible effect on $S$ compared to reef hydrodynamics. The tidal phase-averages of $S$ on the reef platform (Figure 5) demonstrate the strong influence of flow speed and water depth on $S$. Mass-transfer velocities rose sharply during the peak flood and ebb periods (0 – 1.5 and 4 – 6 h after reef flooding, respectively).

The largest $S$ each tidal cycle occurred at the beginning of flood tide, characterized by high flow speeds (~0.5 m s$^{-1}$) and minimum water depths (~0.4 m) on the reef platform (Figure 2); values of $S$ during flood tide were ~30% greater at CR compared to SG, which was due to the larger flow speeds and shallower water depths that occurred near the reef crest (Figure



2). Lowest $S$ each tidal cycle (Figure 5) occurred at high tide when flow speeds became negligible and reef water depths were comparatively large (~2.5 m). Values of $S$ were relatively small (~5 m d$^{-1}$ for NO$_x$) later in ebb tide (8 – 12 hours after reef flooding) and were similar between SG and CR. As $S$ was estimated over a full spring-neap tidal cycle, the ranges of values shown (Figure 5) are from the most (spring) and least (neap) energetic tidal cycles, which differ by a factor of <4.

The mass-transfer-limited nutrient fluxes $J_{MTL}$ were a function of both $S$ as well as the local nutrient concentrations (Eq. 3). Fluxes showed variability over the tidal cycle associated with $S$, but also showed prominent differences between benthic communities and seasons related to nutrient concentrations. Elevated NO$_x$ concentrations at CR (Figure 3a,b) resulted in rising $J_{MTL}$ during the final 6 hours of ebb tide, while low NO$_x$ concentrations at SG resulted in low $J_{MTL}$, especially during ebb tide

10 (Figure 6). Similar concentrations of DIP (Figure 3e,f) and NH$_4^+$ between sites resulted in similar $J_{MTL}$ between CR and SG for both nutrient species (Figure 6c,d). The influence of seasonal changes in offshore nutrient concentrations was evident, particularly for NO$_x$, where elevated levels during Feb (Table 2) resulted in a doubling of $J_{MTL}$ during the first 6 hours of each tidal cycle, compared to Oct (Figure 6a,b). Seasonal differences in $J_{MTL}$ were also found for DIP, where elevated fluxes occurred during Oct (compared to Feb) due to higher DIP concentration in the dry season (Table 2, Figure 6c,d). The maximum

15 potential release of DIN and DIP to the water column assuming uptake was mass-transfer-limited ($J_{release}$, Eq. 7), was calculated for every instance of measured $J_{net}$ (Figure 4). In the case of NO$_x$, $J_{release}$ was roughly double $J_{net}$ (Figure 4a) due to the large NO$_x$ release measured on the reef platform. Whereas for NH$_4^+$ and DIP, $J_{release}$ was on the order of $J_{MTL}$ due to negligible values of $J_{net}$ (Figure 4b,c). Overall mean rates of $J_{MTL}$ and $J_{release}$ for DIN did not show seasonal differences (Table 3), which was likely a function of these estimates only occurring during a portion (ebb) of the tidal cycle.

When $S$ was averaged over individual semidiurnal tidal cycles (e.g., mean of all $S$ within a tidal cycle, beginning with reef flooding), the difference between SG and CR was only ~1 m d$^{-1}$ (Figure 7). Mass-transfer velocities for NO$_x$ and NH$_4^+$ were of similar magnitude over the tidal cycle, while those for DIP were ~50% lower (Figure 7); this was a function of the diffusivity of each of these solutes (Li and Gregory, 1974). When $J_{MTL}$ was similarly averaged over individual tidal cycles (Figure 8),

25 community and seasonal differences in $J_{MTL}$ described previously (Figure 6) were prominent. Uptake of NO$_x$ showed the greatest differences between seasons and sites, with uptakes rates during the wet season greater than dry season rates by a factor of ~2. Similarly, estimates of DIP uptake were slightly enhanced during the dry season compared to wet season rates, while uptake of NH$_4^+$ was similar between seasons and sites (Figure 8).



## 4 Discussion

### 4.1 Oceanic nutrient supply

The measurements of offshore nutrient concentrations presented in Table 2 are among the first published for the Kimberley region (Jones et al., 2014) and are the only (to our knowledge) published record that includes measurements during the wet

season. Concentrations of dissolved nutrients ($NO_x$, $NH_4^+$, DIP, and DON) were at the upper end of typical values in coral reef waters worldwide, especially in the case of DON, which far exceeded the <5 $\mu$M common in reef waters (Atkinson and Falter, 2003). Measurements from the coastal Kimberley (Table 2) also exceeded long-term mean values from inshore waters of the Great Barrier Reef (GBR) during both the wet and dry seasons (Furnas et al., 2005;Schaffelke et al., 2012). The Kimberley region shares similar rainfall patterns, tidal ranges, and low levels of catchment alteration with the northern GBR

(at a similar latitude to the Kimberley), yet concentrations of DIN and DIP measured in this study were an order of magnitude greater than those from the wet tropics (Furnas et al., 2005;Schaffelke et al., 2012). These observations, coupled with elevated concentrations of chlorophyll *a* and particulate nutrients (Gruber et al., 2018) relative to 'typical' oligotrophic reef waters, suggest that some coastal Kimberley reefs may experience naturally mesotrophic conditions.

Wet season terrestrial discharge events deliver sediment and nutrients to coastal waters of northern Australia (Brodie et al., 2010;Devlin and Schaffelke, 2009;Schroeder et al., 2012). Offshore concentrations of $NO_x$ and $NH_4^+$ measured in our study approximately doubled during the Feb field experiment compared to Oct, whereas DIP and DON were similar between seasons (Table 2). Whether this increase is due to river discharge or coastal oceanographic processes is not presently clear in the Kimberley region, and warrants future study. Ratios of offshore DIN:DIP were 4.3 and 10.7 in Oct and Feb, respectively

(Table 2), with the value during Oct similar to the DIN:DIP ratio of ~3:1 previous found in coastal Kimberley waters during the dry season (Jones et al., 2014). These values are below the Redfield ratio (16:1), suggesting that pelagic production may be N-limited. This is common for reef waters generally, although long-term averages of inshore GBR waters are generally <3:1 even during the wet season (Furnas et al., 2005;McKinnon et al., 2013;Schaffelke et al., 2012). This suggests that N-limitation may be less severe in the Kimberley than in GBR waters, particularly during the wet season.

### 4.2 Rates and sources of benthic release of DIN and DIP

Benthic nutrient fluxes measured using the control volume technique ($J_{net}$) showed net release of $NO_x$ on Tallon (Figure 4a), while $NH_4^+$ and DIP fluxes varied between uptake and release (Figure 4b,c) but overall were negligible during the ebb tide (Table 3). Previous studies of reef nutrient fluxes in flumes or other controlled environments have generally shown uptake approaching the limits of mass-transfer for $NH_4^+$ (e.g., Atkinson et al., 1994;Cornelisen and Thomas, 2009;Larned and

Atkinson, 1997;Thomas and Atkinson, 1997), DIP (reviewed in Cuet et al., 2011b), and less frequently for $NO_x$ (e.g., Baird et al., 2004). Yet net release of all three species (especially $NO_x$) clearly occurs in situ as concentrations on many reefs exceed those offshore (e.g., Hatcher and Frith, 1985;Leichter et al., 2013;Rasheed et al., 2002), and release rates up to 20 mmol $NO_x$



$m^{-2}$ $d^{-1}$, 12 mmol $NH_4^+$ $m^{-2}$ $d^{-1}$, and 2 mmol DIP $m^{-2}$ $d^{-1}$ have been measured with in situ studies (Miyajima et al., 2007a;Miyajima et al., 2007b;Silverman et al., 2012;Wyatt et al., 2012). We have not considered nitrogen inputs from other sources such as $N_2$ fixation (Cardini et al., 2014) or reef porewater advection during ebb tide (Santos et al., 2011), which may result in an overestimation of DIN release on Tallon. However, given that $NO_x$ concentrations generally approach detection

limits in reef porewater (Sansone et al., 1990;Tribble et al., 1990) and $N_2$ fixation adds to the $NH_4^+$ pool, it seems unlikely that either of these processes dominate the observed nutrient fluxes.

If we assume that the fluxes discussed above ($J_{net}$) simultaneously occur with uptake of DIN and DIP near the limits of mass-transfer, this gives a gross release ($J_{release}$) of ~10 mmol N $m^{-2}$ $d^{-1}$ and ~0.5 mmol P $m^{-2}$ $d^{-1}$ (Table 3). Previous work has

attributed inorganic nutrient release to remineralization of particulate material by benthic filter-feeders (Ribes et al., 2005;Wyatt et al., 2012) and detritivores (Silverman et al., 2012), which can graze PON on the order of DIN release rates, as well as nitrification by sponge communities (Southwell et al., 2008). In the case of Tallon reef, uptake of phytoplankton (0.95 mmol N and 0.20 mmol P $m^{-2}$ $d^{-1}$) (Gruber et al., 2018) is on the order of $J_{release}$ in the case of P, but is much smaller than $J_{release}$ of N. Large particles (such as entire fronds of macroalgae) are rare but can form a major component of the particulate organic

pool on some reefs (Alldredge et al., 2013); remineralisation of similar material (rather than small particles like phytoplankton) may be the source of the observed DIN release on Tallon. Finally, fluxes of DON on the order of $J_{net}$ were measured on Tallon, with net uptake occurring during the Feb experiment (Figure 4d). The dynamics of DON in reef systems have been addressed in a few studies (e.g., Haas and Wild, 2010;Thibodeau et al., 2013;Ziegler and Benner, 1999), and there is some evidence that reef organisms including corals (Ferrier, 1991), sponges (Rix et al., 2017), and seagrasses (Vonk et al., 2008) can directly

utilise DON. In summary, gross release of DIP may be derived from phytoplankton uptake on Tallon reef, but released DIN exceeds phytoplankton inputs and is likely derived from additional sources including remineralisation of large particles and DON.

**4.3 Tidal and seasonal forcing of mass-transfer-limited fluxes**

Few estimates of nutrient uptake rate $S$ exist for in situ reef communities; the majority of previous estimates come from

controlled flume experiments and are in the range of 2 – 15 m $d^{-1}$ (reviewed in Atkinson and Falter, 2003). Uptake rates are strongly dependent on flow and roughness characteristics (Falter et al., 2016), and in wave-dominated systems $S$ can vary by an order of magnitude across the reef (e.g., from 25 m $d^{-1}$ on the forereef to 5 m $d^{-1}$ in the backreef) as bottom stress from wave forcing declines (Wyatt et al., 2012;Zhang et al., 2011). In wave-dominated systems, $S$ would be expected to be reasonably consistent while offshore wave forcing remain similar (e.g., at scales of days – weeks). Estimates of $S$ from Tallon reef show

uptake rates varying rapidly on the scale of hours or even minutes; for instance, uptake rates for $NO_x$ decreased by an order of magnitude (~30 – 3 m $d^{-1}$) over the period of an hour during flood tide (Figure 6a). When averaged over longer time-scales (i.e., over individual semidiurnal tidal cycles), estimates of $S$ for DIN and DIP (~9 and ~5 m $d^{-1}$, respectively) were similar to the mean of those measured in previous studies and only differed slightly between seagrass and coral reef zones (Figure 7).



Tallon reef platform experiences flows and water depths particular to its geometry and position relative to mean sea level; therefore, $S$ (and accordingly nutrient uptake) will vary in other tide-dominated reef communities as a function of these factors.

Estimates of mass-transfer-limited uptake of DIN and DIP varied over a tidal cycle with $S$, but also showed differences in uptake with reef zone and season (Figure 6). Reef zones were similar in DIP uptake rates, but rising concentrations of $NO_x$ in the coral zone during ebb tide caused estimates of $J_{MTL}$ to increase compared to the seagrass zone (Figure 6a,b). Previous work on Tallon reef has shown that the coral zone is ~20% more productive than the seagrass zone (Gruber et al., 2017), which may be related to this difference in potential nitrate fluxes. Concentrations of $NO_x$ and $NH_4^+$ were elevated in the wet season, while DIP declined compared to the dry season (Table 2); these seasonal differences were evident in the mass-transfer-limited nutrient fluxes even when integrated over individual semidiurnal tidal cycles (Figure 8). Ratios of DIN:DIP mass-transfer-limited uptake during Oct were 8.6 and 10.8 for seagrass and coral zones, respectively (Figure 8). These ratios are well below the tissue N:P ratio of 30:1 typical of reef primary producers (Atkinson and Smith, 1983) and suggest that producers on Tallon reef may be strongly N-limited (at least during the dry season). This is supported by low N:P ratios (14:1) measured in *Thalassia* leaf tissue from Tallon reef during Oct (N. Cayabyab, unpubl.). During Feb, ratios of DIN:DIP mass-transfer-limited uptake were 21.5 and 21.3 for seagrass and macroalgal zones, respectively (Figure 8), which suggests that N-limitation may be somewhat alleviated due to increases in oceanic DIN during the wet season.

### 4.4 Comparison of wave and tidal forcing

This study suggests several important differences between wave- and tide-dominated reef biogeochemistry, which are controlled by the hydrodynamic regime. Firstly, the 'source' of a water parcel overlying a particular benthic community differs between wave- and tide-dominated systems. In a simplified wave-driven reef, offshore (oceanic) water moves from reef crest to back reef roughly unidirectionally. Thus, benthic communities are subjected to the physico-chemical water properties present in offshore waters modified by the communities 'upstream' of them. In a simplified tide-driven reef, flow direction changes throughout the tidal cycle; during flood tide, offshore waters enter the reef, while during ebb tide, waters from the backreef traverse all 'downstream' communities. These flow patterns control water residence times within the reef community. In wave-dominated reefs, flow speeds are driven by wave-breaking on the reef, creating residence times on the scale of ~hours; wave energy can be generally consistent in time over ~days – weeks (Lowe and Falter, 2015). In tide-dominated systems, reef waters exchange with offshore waters at timescales ≤ a semidiurnal (or diurnal) tidal cycle; this residence time will vary depending on the reef vertical position relative to mean sea level and its morphology. Finally, there are marked differences in nutrient uptake rates between wave- and tide-dominated reefs. The consistency of wave energy at scales of ~days – weeks likely drives similarly consistent mass-transfer-limited nutrient uptakes rates on wave-dominated reefs. On reefs with strong tidal forcing however, flow speeds are highly variable throughout the tidal cycle and mass-transfer-limited uptake can vary by an order of magnitude within ~hours - minutes. Flow speeds also change over the spring-neap tidal cycle (~15 days); on Tallon reef, mass-transfer-limited uptake rates were ~2-4 fold greater during spring tides relative to neap tides. Further work on

tidally-forced reefs is necessary to understand the morphological and hydrodynamic properties that distinguish them from wave-driven reef systems. Such studies will improve predictions of reef water temperatures (and coral bleaching), in situ calcification rates, and many other physically-linked biological processes that determine the health and resilience of coral reef communities.

## 5   Conclusions

In conclusion, this study was one of the first to measure rates of in situ benthic nutrient uptake and release on a tidally-forced reef. We found that reef communities released a moderate amount of DIN, potentially derived from the remineralization of phytoplankton, large organic material, and DON. The strong tidal forcing of this reef drives large variability (an order of magnitude) in mass-transfer-limited nutrient uptake rates at short time scales (minutes – hours), and uptake can be enhanced

in reef zones downstream of where DIN release occurs. Tallon reef displays some indications of nitrogen-limitation during the dry season, which may be relieved during the wet season; seasonal increases in offshore nitrate concentrations increased mass-transfer-limited uptake rates by a factor of ~2. This work identifies some hydrodynamic properties of tide-dominated reefs that control their biogeochemistry and help define them in comparison to wave-dominated reefs.

### Author contribution

Field experiments were designed by RKG, RJL, and JLF. Fieldwork was conducted by RKG and RJL. RKG analysed the results and prepared the manuscript with contributions from RJL and JLF.

### Competing interests

The authors declare that they have no conflict of interest.

### Acknowledgements

This work was conducted on Bardi Jawi sea country and we acknowledge the Traditional Owners past, present, and emerging who care for this country. We thank the Bardi Jawi Rangers and Kimberley Marine Research Station staff for providing assistance and local knowledge during field experiments. We thank Michael Cuttler, Jordan Iles, Miela Kolomaznik, and Leonardo Ruiz-Montoya for helping with fieldwork. Funding was provided by the Western Australian Marine Science Institution's Kimberley Marine Research Program (Project 2.2.3), an Australian Research Council Future Fellowship

(FT110100201) to RJL, and the ARC Centre of Excellence for Coral Reef Studies (CE140100020).



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





**Figure captions**

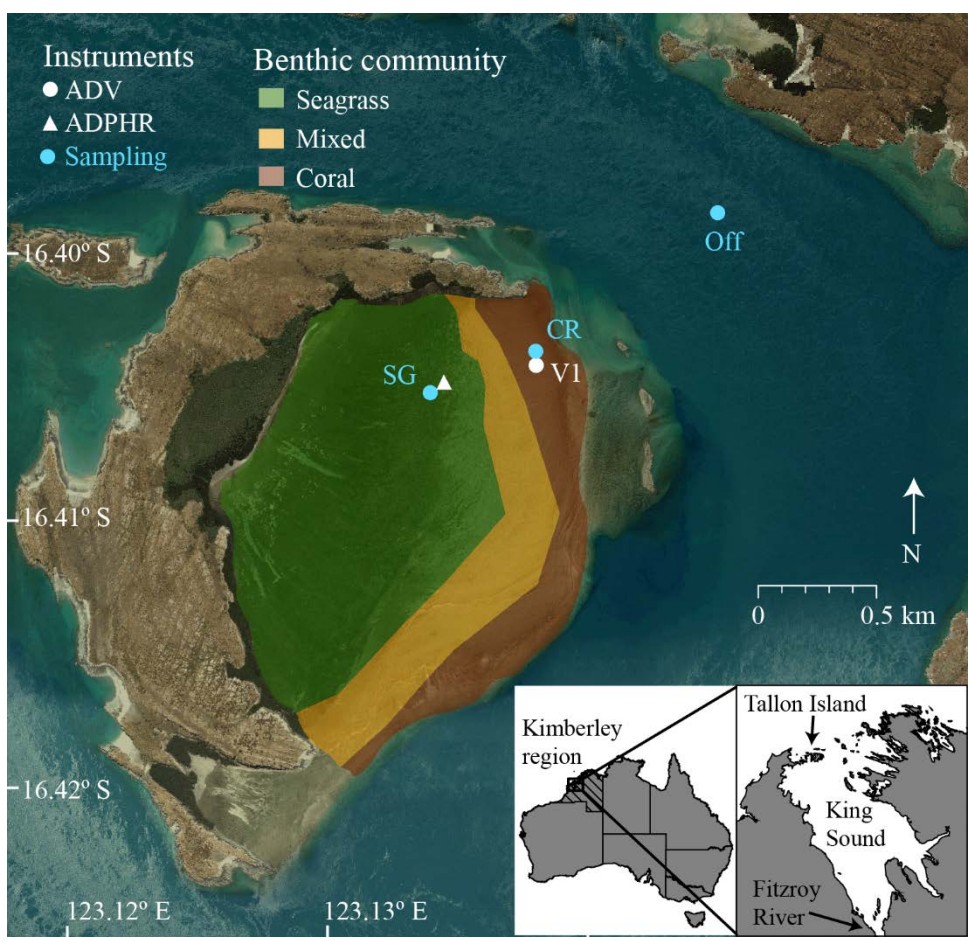

Figure 1. Deployment locations of hydrodynamic instrumentation and water sampling locations on Tallon reef platform and offshore. Inset shows Tallon Island location in the west Kimberley region of Australia. ADV refers to acoustic Doppler velocimeters and ADPHR refers to acoustic Doppler profiler.





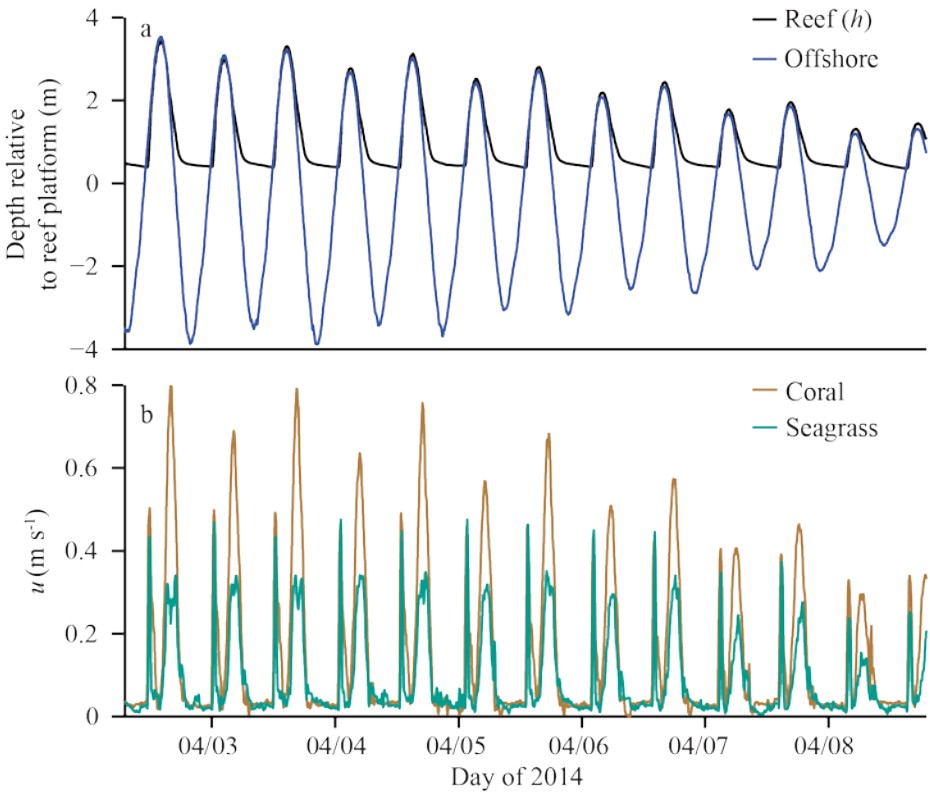

**Figure 2.** Selected time-series of spring-neap transition showing a) water depths on the reef (*h*) and offshore, with depth-averaged flow speed *u* in b) coral and seagrass-dominated zones.







**Figure 3. Measurements of a,b) nitrate (NO$_x$), c,d) ammonium (NH$_4^+$), and e,f) dissolved inorganic phosphorus (DIP) from water samples during Oct (left column) and Feb (right column) field experiments. Samples were taken at two reef stations [coral (CR) and seagrass (SG) dominated zones] and mean offshore nutrient concentrations are shown (blue dashed line). Tidal phase-averaged water depth $h$ is also shown (black line).**







**Figure 4. Fluxes of a) nitrate (NO$_x$), b) ammonium (NH$_4^+$), c) dissolved inorganic phosphorus (DIP), and d) dissolved organic nitrogen (DON) along the study transect during both field experiments. Net benthic fluxes ($J_{net}$) were estimated using the CoVo approach, while mass-transfer-limited uptake ($J_{MTL}$) was calculated from reef platform flow and nutrient concentrations, and nutrient release ($J_{release}$) was estimated from net and MTL fluxes.**





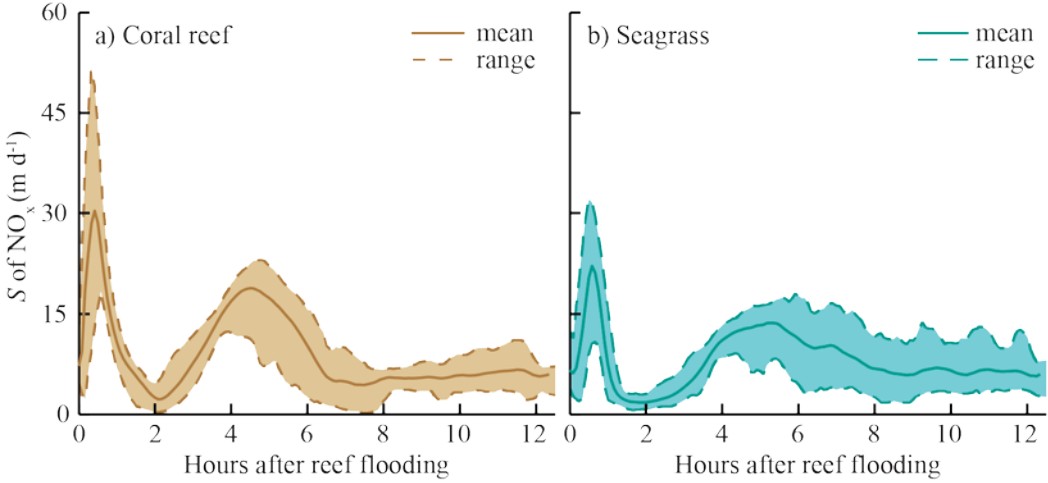

**Figure 5. Tidal phase-averaged mass-transfer velocity $S$ for nitrate ($NO_x$) in a) coral (CR) and b) seagrass (SG)-dominated zones. Phase-averages are the mean of all measurements occurring at the same point in the tidal cycle (i.e., mean of all $S$ at high tide), and the range represents conditions during spring and neap tidal cycles.**





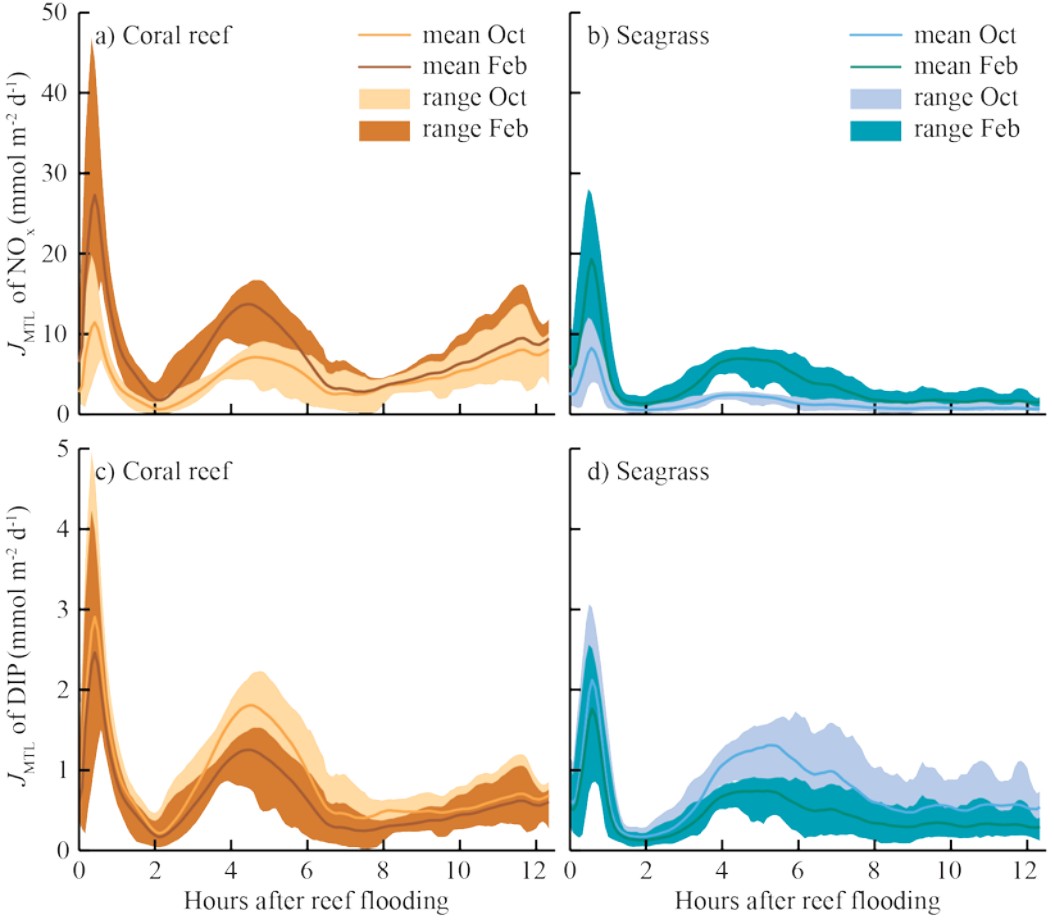

**Figure 6.** Tidal phase-averaged mass-transfer-limited uptake rates of $J_{MTL}$ for a,b) $NO_x$ and c,d) DIP in both coral and seagrass-dominated zones over a full spring-neap cycle. Phase-averages are the mean of all measurements occurring at the same point in the tidal cycle (i.e., mean of all $J_{MTL}$ at high tide). Shaded areas of $J_{MTL}$ indicate the range, where maximum values approximate uptake during spring tides and minimum values during neap tides. Estimates of $J_{MTL}$ were calculated using tidal phase-averaged nutrient concentrations from Oct and Feb field experiments (Figure 3) and mass-transfer velocity $S$ (Figure 5).





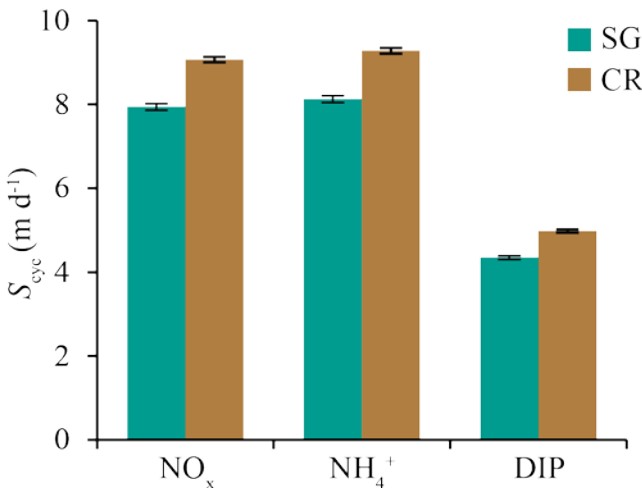

**Figure 7. Means (± standard deviation) of mass-transfer velocity *S* for all individual semidiurnal tidal cycles (*n* = 23) for nitrate (NO$_x$), ammonium (NH$_4^+$), and dissolved inorganic phosphorus (DIP). Values are from seagrass (SG) and coral (CR)-dominated communities.**



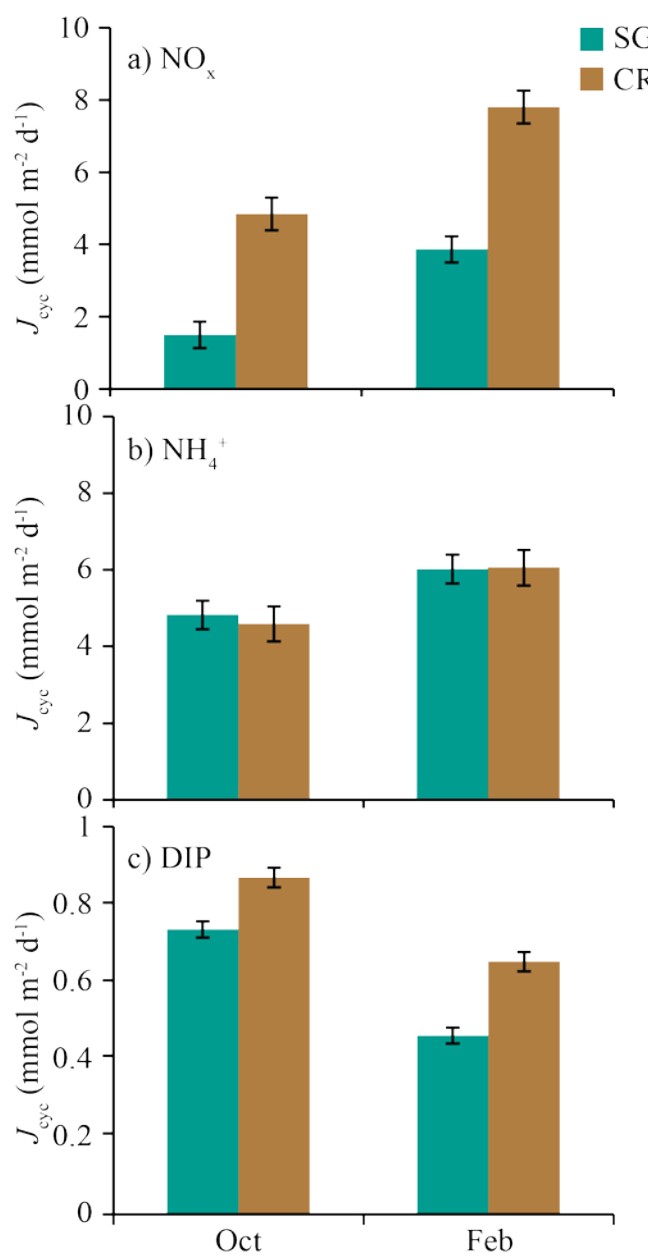

**Figure 8.** Means (± standard deviation) of mass-transfer-limited uptake $J_{\text{MTL}}$ for all individual semidiurnal tidal cycles ($n = 23$) for a) nitrate ($NO_x$), b) ammonium ($NH_4^+$), and c) dissolved inorganic phosphorus (DIP). Values are from seagrass (SG) and coral (CR)-dominated communities during Oct and Feb field experiments.





**Tables**

**Table 1. Number of duplicate water samples collected during both field experiments at offshore (Off), coral (CR), and seagrass (SG)-dominated sites for analysis of dissolved inorganic nitrogen (DIN), dissolved inorganic phosphorus (DIP), and dissolved organic nitrogen (DON).**

| Dates | Season | Sites | DIN/DIP | DON |
|---|---|---|---|---|
| | | Off | 26 | 26 |
| 5 - 20 Oct 2013 | Dry | CR | 36 | 36 |
| | | SG | 33 | 32 |
| | | Off | 15 | 15 |
| 4 - 9 Feb 2014 | Wet | CR | 14 | 14 |
| | | SG | 13 | 13 |

**Table 2. Summary of mean (*standard deviation*) conditions in offshore waters (Off) during Oct and Feb field experiments. Number of samples represented by each mean is shown in Table 1. Nutrient species measured are nitrate/nitrite ($NO_x$), ammonium ($NH_4^+$), dissolved inorganic phosphorus (DIP), and dissolved organic nitrogen (DON).**

| | Tide range (m)[†] | | | | Concentration ($\mu$M) | | | |
|---|---|---|---|---|---|---|---|---|
| | Spring | Neap | Salinity | Temp (°C) | $NO_x$ | $NH_4^+$ | DIP | DON |
| Oct 2013 | 6.7 | 2.6 | 34.7 (*0.02*) | 27.8 (*0.29*) | 0.40 (*0.09*) | 0.37 (*0.12*) | 0.18 (*0.02*) | 12.7 (*2.4*) |
| Feb 2014 | 7.0 | 2.1 | 34.2 (*0.06*) | 30.1 (*0.06*) | 0.92 (*0.19*) | 0.69 (*0.23*) | 0.15 (*0.03*) | 10.7 (*2.5*) |

[†]Difference between max and min water levels



**Table 3.** Mean (*standard error*) net fluxes (in mmol m$^{-2}$ d$^{-1}$) of nutrients determined by the CoVo approach during the Oct and Feb field experiments. Nutrient species include nitrate/nitrite ($NO_x$), ammonium ($NH_4^+$), dissolved inorganic phosphorus (DIP), and dissolved organic nitrogen (DON). Mean net ($J_{net}$), mass-transfer-limited ($J_{MTL}$), and release ($J_{release}$) fluxes are from samples taken during the final 6 hours of ebb tide and do not represent fluxes at all phases of the semidiurnal tidal cycle.

|  |  | Oct 2013 | Feb 2014 |
|---|---|---|---|
| $NO_x$ | $J_{net}$ | -2.3 (*0.29*) | -2.4 (*0.47*) |
|  | $J_{MTL}$ | 3.3 (*0.04*) | 3.1 (*0.20*) |
|  | $J_{release}$ | -5.5 (*0.30*) | -5.5 (*0.47*) |
| $NH_4^+$ | $J_{net}$ | -0.06 (*0.07*) | -0.6 (*0.11*) |
|  | $J_{MTL}$ | 4.6 (*0.05*) | 4.5 (*0.09*) |
|  | $J_{release}$ | -4.6 (*0.09*) | -5.0 (*0.14*) |
| DIP | $J_{net}$ | -0.03 (*0.013*) | -0.08 (*0.038*) |
|  | $J_{MTL}$ | 0.56 (*0.006*) | 0.35 (*0.007*) |
|  | $J_{release}$ | -0.59 (*0.014*) | -0.43 (*0.039*) |
| DON | $J_{net}$ | -1.6 (*2.07*) | 3.4 (*1.50*) |

