# Peer review of "Tidal and seasonal forcing of dissolved nutrient fluxes in reef communities"

_Biogeosciences, 2018_

## Referee Comment (RC1) · Anonymous Referee #1 · 6 Nov 2018

Gruber et al. present measurements of nutrient concentrations and dissolved nutrient flux rates from a macro tidal reef in the Kimberley region of Australia. They compare net rates of dissolved nutrient uptake to the theoretical mass transfer-limited uptake rates. They conclude that the reef acts as a source of DIN and DIP to the water column. I find this manuscript well-written with good logic and structure. I have some comments to help improve the readability, especially for readers unfamiliar with the sequences of papers that have come from this group of authors on reef physics-biogeochemistry at this site. Overall, I would classify these comments as minor-to-moderate.

I do have one over-arching question/comment: Tallon reef, and the macrotidal Kimberley, seem like end-members on the spectrum of hydrodynamic conditions experienced on reefs. The authors do a good job referencing the Lowe and Falter (2015) paper highlighting the ubiquity of tidal-dominated reefs (even though most research has gone into studying wave-dominated systems), but how applicable do you think the results here are to other systems? Not many other systems feature large amounts of aerial exposure, asymmetric phase duration, and the massive velocities associated with drainage of the platform. How do these conditions affect the applicability of Tallon reef as a model biogeochemical system whose results can be generalized?

Here are some more detailed comments broken down by sections of the paper.

Abstract: L15-"moderate amount", replace with an actual quantity of nitrate

Introduction: I recommend the authors reverse the ordering of their discussion of organic and inorganic nutrients. Since the manuscript focuses on DIN and DIP, it stands to reason that they should be discussed before the refractory DON and DOP pools. I recommend that you move the paragraph discussion inorganic nutrients (p. 2, L 10-23) ahead of the discussion of organic nutrients (p. 2, L 1-8).

Methods:

Section 2.1: Please insert your well-worded definition of "tidal phase-averaging" from the Fig. 5 caption into p. 4, L 10-13. The current wording in this paragraph is ambiguous about whether data are averaged within the phases of a tidal cycle or across tidal cycles. The definition in the figure caption relieves this ambiguity.

Section 2.2. How were water samples collected exactly? Did you hold the syringe at the surface and draw up the water? Or was it just above the benthos? Did you directly collect the water with the syringe? Or did you use some auto-sampler to sample the water, and then draw up into a syringe? A few more details would be helpful for conveying you're sampling plan to readers.

Section 2.3 As it stands now, I think this paragraph could benefit for a few clarifications. First, I think the authors need to be more explicit that the J_net estimate is not for the seagrass site, nor for the coral site, but is the average flux rate along the transect mov-

ing from the seagrass to the coral site. Even though this may be obvious for people experienced in control volume approaches, I think it is less intuitive for people without control volume backgrounds. When the authors write "C_bar is the mean of concentrations at both stations", do you mean that you average C_bar between the seagrass and coral site at each time-step? If so, say so. I find the current wording confusing. Please either provide more explanation for why you use "local benthic flux" to describe the unsteadiness term on the RHS of Eq. 2. At the end of this paragraph, the authors state ". . . this method is described in greater detail in Gruber et al. (2017).". Is this in reference to your interpolation of advective estimates to when you have nutrient samples? I think the authors need to be clear about how the interpolation proceeds, and do so in a manner which does not require reading Gruber et al. (2017) to understand the interpolation.

Section 2.4 There is no equation for C_D (p. 6, L 3-5). Instead the authors state ". . .following the same approach as used in estimates of reef metabolism (Gruber et al. 2017).". Please give additional information on C_D so that interested readers could evaluate your C_D model without having to read Gruber et al. (2017).

Minor comments for the Methods: Please add in some information about the precision of your nutrient measurements.

Please list Sc value numbers for your inorganic nutrients (or at least diffusion coefficients) (p. 6, L 1-2).

Please quantify all error terms that went into your Monte Carlo simulations (p. 6, L 30)

Results: p. 7, L 26: Quantify changes in S due to diffusivity p. 7, L 27: Quantify temperature effects on S (don't need a lot here, but something to give readers a sense if the error from ignoring temperature variability is on the order of 0.01%, 0.1%, 1%, 10%, etc. would be useful).

Figures: Fig 1: Do you have example photos from the SG and CR sites that could be

added to this figure to help convey the communities described in the 1st paragraph of the Methods? I think this visual representation would help readers understand the two sites. Fig. 4: Please describe the error bars (e.g., SD, SE, 95% CI, etc.) Fig. 5: I think it would be interesting to put dashed lines on these plots to show the range of S estimates from flume and wave-driven field system studies (p. 10, L 24-28). These would really help show that the variability in S in tidal-dominated systems is far larger than in previously studied systems. Figs. 5 and 6: I think these two figures should be combined, and it would really help readers to see them as a multi-panel plot so that they can understand how closely the J_MTL estimates mirror the S estimates (or alternatively, depart from each other).

Tables: Table 1: "Number of duplicate samples"- does this mean the total number of replicates analyzed? Or the total number of unknown water samples collected, each of which were duplicated? Please clarify. Tables 1 and 2: I think these tables can be combined. This would streamline the manuscript by reducing the number of tables (as it stands right now, Table 1 adds little unique information).

Grammar/typos: p. 1, L 30: Correct subscripting/superscripting of NOx and NH4+ p. 7, L 24: "through" (though)

---

## Referee Comment (RC3) · Anonymous Referee #2 · 14 Jan 2019

This MS reports fluxes of dissolved inorganic nitrogen and phosphorus and theoretical mass-transfer-limited uptake rates on a strongly tide-dominated reef platform. The amount of nutrients that is released in the water column is calculated from these two data sets.

General evaluation:

Overall, this is a very interesting paper, showing that mass-transfer-limited uptake rates may vary by an order of magnitude on the scale of minutes to hours on tide-dominated reefs, due to substantial variability in flow speeds and water depths over the tidal cycle. Differences between wave- and tide-dominated reef biogeochemistry that are due to the hydrodynamic regime are nicely highlighted. I have a number of relatively minor comments aimed at clarifying the methods and results (detailed below), which the au-

[Figure]

thors should be able to answer easily. The main concern is that nutrient concentrations were not measured in Feb during the first 6 hours of the tidal cycle. Missing data were replaced by nutrient concentrations in offshore waters, which is maybe not perfectly supported by the data, particularly with regard to NOX (Figure 3). I would therefore recommend more caution in the conclusions regarding seasonal differences in JMTL (see below). The discussion about the implications in terms of nutrient limitation is worthwhile, however.

Comments in detail:

Introduction

p. 2 Line 5: you could add a reference about corals: Grover, R., Maguer, J. F., Allemand, D., & Ferrier-Pagès, C. (2008). Uptake of dissolved free amino acids by the scleractinian coral Stylophora pistillata. Journal of Experimental Biology, 211(6), 860-865.

Methods

2.1 Field site

p. 4 Lines 8-10: The hydrodynamic study of Lowe at al. (2015) was performed in March-April 2014, while the MS reports nutrient concentrations from October 2013 and February 2014. Although this is justified later in the MS, the reason why hydrodynamic data collected at the same time as nutrients were not used to calculate mass-transfer-limited uptake rates is unclear at this point.

p. 4 Lines 11-13: It is unclear if this concerns mass-transfer-limited uptake rates only, or nutrient fluxes as well.

2.3 Control volume approach

Line 8: "Depth-averaged flow speeds were bin-averaged": do you mean that you first averaged flow speeds in each bin (at 5 min intervals) before depth averaging?

Lines 13-21: I think this paragraph needs some clarification. You are using the mean of concentrations at both stations to calculate the local benthic flux, but you explain then that this local term represents nutrient uptake or release occurring at a sampling station (in my mind, CR or SG, while you are calculating Jnet on the transect). The advective flux is described as nutrient uptake or release during transit between sampling stations, which I also find very confusing. Should not it be what is added or subtracted to the transect due to water transport? The minus sign in front of Jnet is also surprising at first glance. It might be more understandable to state that the sign of Jnet was reversed so that uptake is positive and release negative.

2.4 Uptake rates at the limits of mass-transfer

p.5 Lines 25-30: In the results, JMTL is first calculated for both CR and SG (Figure 6), not along the study transect. Maybe the reasoning would be easier to follow if there was first a paragraph about the calculation of mass-transfer velocity and JMTL at each one of the two stations, and then, a new paragraph explaining the calculation of Jrelease.

p.5 Line 30 to p.6 Line 9: This, although necessary, is really hard to follow. There are very nice explanations given in the results (p. 7 Lines 23-27) that might help the reader to go through the equations. Would it be possible to integrate these explanations into this paragraph, specifying the parameters whose variability has the most influence? Maybe, in a second step, you could simplify equation 4, as some parameters are constants (or nearly constant). This would highlight the influence of flow speed (and possibly water depth: the drag coefficient CD was taken as 0.02 in Gruber et al. (2017), but I didn't understand if this is the case here) and nutrient diffusivity?

In Gruber et al. (2017), u* is a function of ux, not u (this MS). Is there an explanation? I would also suggest that you explain why you are using u in S calculation, and not ux.

Lines 16-28: I would suggest to give these informations before Line 10 (calculation of Jrelease). Could you please clarify which instruments were deployed exactly? On Figure 1, a velocimeter is shown at CR, but you are not using the data, right? Jnet was

calculated from the ADCP data at SG, are you using the same current speed data for the calculation of JMTL and then Jrelease, or the data from Lowe et al. (2015)?

Lines 29-30: This deserves more explanations. Do you mean, for example, that error bars on Figure 4 are uncertainties in estimates of each one of the calculated fluxes? Please add this information in the caption.

Results

3.1 Nutrient concentrations and measured fluxes

Line 4: From Table 2, water temperature is about 2°C warmer in Feb (not 3°C).

Some observations could be supported by statistics: line 6 (DIP and DON are slightly lower in Feb); lines 6-7 (concentrations are similar on the reef and offshore until 6 hours after flooding); lines 18-19 (no difference between Oct and Feb for NH4?).

3.2 Mass-transfer velocity and nutrient uptake

It is unclear from p. 6 lines 3-5 that the drag coefficient was calculated as a function of water depth to draw Figure 5 (the drag coefficient was taken as 0.02 in Gruber et al., 2017). Please clarify in the Methods.

p. 7 Lines 26-27: Can you roughly quantify the effect of temperature on S?

p. 7 Line 28: Could you add on Figure 5 the drag coefficient as a function of hours after reef flooding (and maybe flow speed as well, to avoid having to go back to Figure 2). Also state in the caption that hydrodynamic data (and presumably water depth?) are from March-April 2014 (Lowe et al., 2015), while temperature and salinity from Oct 2013 and Feb 2014 (is that right?).

p. 7 Line 32: Figure 2 doesn't show water depth at SG and CR. Could you add tidal phase-averaged water depth at each site on Figure 3?

p. 8 Line 4: Could you explain "which differ by a factor of <4"? Is it the ratio of the flow

[Figure]

speeds?

Figure 6: There are missing nutrient data, especially in Feb (0-6 hours after reef flooding; Figure 3). I understand from p. 6 Lines 24-26 that missing data were replaced by nutrient concentrations in offshore waters. From Figure 3, this looks acceptable in Oct, but maybe less in Feb, especially for NOx. Could you show on Figure 6 the time periods during which nutrients were actually measured?

p. 8 Line 10: JMTL is not shown for ammonia. Why?

p. 8 Lines 11-13: This looks speculative, as nutrient concentrations were not measured in Feb during the first 6 hours of the tidal cycle (see previous comment on Figure 6). Please state clearly that you are assuming that NOx concentrations are similar on the reef and offshore during this period and add some comment in the Discussion.

Figure 4: I assume that JMTL is the mean of the values shown on Figure 6 for SG and CR?

p. 8 Line 17: Is "NOx release" "net NOx release"?

Seasonal differences: could you add your stats as a column in Table 3?

p. 8 Line 19: This comment refers to Lines 11-13 (see above). I would suggest to simply state that, contrary to DIP, DIN concentrations are similar at both seasons during the part of the tidal cycle studied.

P 8 from Line 21, and Figures 7 and 8: I understand that you averaged S and JMTL over each one of the 12-hours period available, and then averaged these averages. If that's right, first you should explain why, and then, I don't think averaging averages is the best way to assess standard deviations (they also appear very small in Figure 7, given the range in Figure 5).

Lines 21-24 and Figure 7: I'm not sure this is very useful. The two points about S are: (1) the small difference between SG and CR, which is already described p. 7 Lines

31-32 and p. 8 Line 3, and (2) the difference between DIN and DIP which you can easily talk about after p. 7 Lines 25-26.

From Line 24 and Figure 8: Again, I would not recommend using all data from Figure 6 due to missing nutrient values. Differences between seasons and sites are already described (p. 8 Lines 6-14).

Discussion

p. 9 Line 17 (DIP and DON): DIP and DON are "slightly lower (. . .) in Feb" p. 7 Line 6 and "similar between seasons" p. 9 Line 17. One of these two sentences needs to be re-written after stats are performed.

Line 28 to p.10 Line 2: Do you mean that mass-transfer-limited uptake was demonstrated in controlled environments because nutrient release was negligible compared to uptake?

p. 10 Line 4: overestimation of DIN release on Tallon: I don't understand your point. Whatever the source, I think that your calculation of DIN release is fine. Could you clarify?

References

There are two papers by Lowe et al., 2015. Please use 2015a and 2015b throughout the text.

---

## Referee Comment (RC4) · Anonymous Referee #3 · 16 Jan 2019

The study reports results from measurements of dissolved nitrogen and phosphorus fluxes in a tide-dominated reef. The fluxes were estimated using a one-dimensional control volume method, i.e. nutrient concentrations were measured on the reef platform at two points along the flow path of the tidal currents, and from the concentrations changes occurring between the two points, fluxes were estimated. The measurements suggest a release of nitrate, while fluxes of NH4+ and DIP varied between net uptake and release.

The results are interesting and produce new insights into the functioning of tide-dominated reefs. A few points need clarification and revision:

This is a subtropical environment with healthy seagrass and coral cover. Nutrients in the water column are low and it is surprising that the reef platform with seagrass and

healthy coral releases nitrate during low tide. Assuming that during the low water phase photosynthesis of vascular and unicellular plants reaches a maximum, one would expect nutrient uptake of the benthic community resulting in a net in uptake. Tidal current vector fields previously published by the authors for this study site (Lowe et al. 2015) revealed that the tidal water in and outflow is not symmetrical, resulting in a residual current entering the reef at its southeastern edge and leaving at its northeastern edge. These residual currents include flows parallel to the reef, which may cause that the two sampling stations measure some waters that took different pathways over the reef platform. It is conceivable that the residual currents transport water with different nutrient concentrations to the two measuring stations and that the observed difference in nitrate concentration between the two stations is a function of the pathways of the residual currents. The reef lagoon appears to be lined by mangrove forest and the seagrass community accumulates organic-rich sediments. The mixed zone between seagrass and coral has pockets of sediment and a porous structure. Could release of fluid from the sediments and the porous structure of the mixed zone explain the nutrient increase during decreasing water level as the path of the residual currents is along the mixed zone? This should be clarified.

The tidal range exceeding 8 m is unusual. This is not a typical scenario, which needs to be considered and pointed out when generalizing the results.

Minor comments.

P1L9 a "forcing" is not a "regime", please rephrase.

P1L25 "Reef waters have carbon concentrations that are orders of magnitude greater than nitrogen (N) and phosphorus (P)". If a ratio close to Redfield is applicable here, C concentrations order(s) magnitudes higher than those of the nutrients can be expected. Benthic communities would not be nutrient limited. If carbon is way higher, please rephrase.

P2L10 "labile dissolved inorganic species". "Labile" here seems the wrong word as

some of these inorganic species can be very stable in the marine environment.

P216 "turbulent transport" should be added to the list of the controls of nutrient transfer

P4L8 "reef benthos, which represent the net uptake or release of nutrients". Please add information on nutrient uptake/release of water column organisms

P4L32 "All nutrient concentrations presented are the mean of duplicate samples" why didn't you take triplicate samples, which would have allowed calculation of standard deviation, opening up other options for statistical analysis?

P5L8 How large is the error introduced by using depth-averaged current velocity instead

P11L8 If the coral zone is ~20% more productive than the seagrass zone (Gruber et al., 2017), one would expect an increasing N consumption during decreasing water level as light intensity at the reef surface increases, with higher N demands in the coral zone. The results suggest the opposite, how is this explained?

P11L20 "In a simplified wave-driven reef, offshore (oceanic) water moves from reef crest to back reef roughly unidirectionally. Thus, benthic communities are subjected to the physico-chemical water properties present in offshore waters modified by the communities 'upstream' of them." This should be explained in more detail, as water transported into the reef also has to leave the reef, irrespective of the transport process. This release may traverse the communities that contacted this water before as in the tidal dominated reef.

---

## Author Comment (AC1) · 4 Apr 2019

MS No.: bg-2018-413 Anonymous Referee #1 Reviewer comments are included below denoted as "Ref1". Author response to each comment is given below the comment denoted as "Authors". Please note that page/line numbers correspond to the revised version of the manuscript rather than the original version.

Ref1: Gruber et al. present measurements of nutrient concentrations and dissolved nutrient flux rates from a macro tidal reef in the Kimberley region of Australia. They compare net rates of dissolved nutrient uptake to the theoretical mass transfer-limited uptake rates. They conclude that the reef acts as a source of DIN and DIP to the water column. I find this manuscript well-written with good logic and structure. I have

some comments to help improve the readability, especially for readers unfamiliar with the sequences of papers that have come from this group of authors on reef physics-biogeochemistry at this site. Overall, I would classify these comments as minor-to-moderate.

Authors: We thank the reviewer for their supportive and constructive comments on this manuscript (ms).

Ref1: I do have one over-arching question/comment: Tallon reef, and the macrotidal Kimberley, seem like end-members on the spectrum of hydrodynamic conditions experienced on reefs. The authors do a good job referencing the Lowe and Falter (2015) paper highlighting the ubiquity of tidal-dominated reefs (even though most research has gone into studying wave-dominated systems), but how applicable do you think the results here are to other systems? Not many other systems feature large amounts of aerial exposure, asymmetric phase duration, and the massive velocities associated with drainage of the platform. How do these conditions affect the applicability of Tallon reef as a model biogeochemical system whose results can be generalized?

Authors: The tidal range at Tallon reef ($\sim$8 m) is typical of ranges experienced by other Kimberley reefs, and our results are likely fairly representative of the $\sim$2000 km2 of total reef area in this region. An $\sim$8 m tidal range is indeed extreme compared to most reefs worldwide; however, tidal forcing acting as the dominant process in reef circulation is quite common (for example, most of the Great Barrier Reef has tide-dominated circulation). While the large degree of variability in benthic fluxes presented in this paper would not necessarily be representative of most reef systems, the patterns caused by tides are likely to be common among tide-dominated reefs ($\sim$30% of reefs worldwide). We have added some text to Discussion Section 4.4 to clarify this, as multiple reviewers had similar comments.

Here are some more detailed comments broken down by sections of the paper.

Abstract: Ref1: L15-"moderate amount", replace with an actual quantity of nitrate

Authors: We have added this value to the Abstract.

Introduction: Ref1: I recommend the authors reverse the ordering of their discussion of organic and inorganic nutrients. Since the manuscript focuses on DIN and DIP, it stands to reason that they should be discussed before the refractory DON and DOP pools. I recommend that you move the paragraph discussion inorganic nutrients (p. 2, L 10-23) ahead of the discussion of organic nutrients (p. 2, L 1-8).

Authors: We have changed the order of these paragraphs.

Methods: Section 2.1: Ref1: Please insert your well-worded definition of "tidal phase-averaging" from the Fig. 5 caption into p. 4, L 10-13. The current wording in this paragraph is ambiguous about whether data are averaged within the phases of a tidal cycle or across tidal cycles. The definition in the figure caption relieves this ambiguity.

Authors: We agree this wording was ambiguous and have revised that sentence for clarity.

Section 2.2. Ref 1: How were water samples collected exactly? Did you hold the syringe at the surface and draw up the water? Or was it just above the benthos? Did you directly collect the water with the syringe? Or did you use some auto-sampler to sample the water, and then draw up into a syringe? A few more details would be helpful for conveying you're sampling plan to readers. Authors: We have added some wording to make Section 2.2 p4 L25-26 clearer.

Section 2.3 Ref1: As it stands now, I think this paragraph could benefit for a few clarifications. First, I think the authors need to be more explicit that the J_net estimate is not for the seagrass site, nor for the coral site, but is the average flux rate along the transect moving from the seagrass to the coral site. Even though this may be obvious for people experienced in control volume approaches, I think it is less intuitive for people without control volume backgrounds.

Authors: We agree with the reviewer and have added a sentence to p5 L19 to clarify

this point.

Ref1: When the authors write "C_bar is the mean of concentrations at both stations", do you mean that you average C_bar between the seagrass and coral site at each time-step? If so, say so. I find the current wording confusing.

Authors: We agree with the reviewer and have added text to p5 L17 to clarify.

Ref1: Please either provide more explanation for why you use "local benthic flux" to describe the unsteadiness term on the RHS of Eq. 2.

Authors: This is a naming convention that illustrates we are working in the frame of reference of the sampling stations. We have modified some text to p5 L21 to clarify.

Ref1: At the end of this paragraph, the authors state "... this method is described in greater detail in Gruber et al. (2017).". Is this in reference to your interpolation of advective estimates to when you have nutrient samples? I think the authors need to be clear about how the interpolation proceeds, and do so in a manner which does not require reading Gruber et al. (2017) to understand the interpolation.

Authors: We agree and have added an explanation that clarifies this (p5 L23-27).

Section 2.4 Ref1: There is no equation for C_D (p. 6, L 3-5). Instead the authors state "...following the same approach as used in estimates of reef metabolism (Gruber et al. 2017).". Please give additional information on C_D so that interested readers could evaluate your C_D model without having to read Gruber et al. (2017).

Authors: This was an omission and we have added an equation and explanation for calculating CD (p6 L8-10). It actually wasn't the same method as Gruber et al. 2017 as we found a more realistic (we think) way of estimating the drag coefficient during the time between that article's publication and the submission of this ms.

Minor comments for the Methods: Ref1: Please add in some information about the pre-cision of your nutrient measurements. Please list Sc value numbers for your inorganic
nutrients (or at least diffusion coefficients) (p. 6, L 1-2).

Authors: We have added values for diffusion coefficients to p6 L5-6.

Ref1: Please quantify all error terms that went into your Monte Carlo simulations (p. 6, L 30)

Authors: We have added these terms to p7 L6.

Results: Ref1: p. 7, L 26: Quantify changes in S due to diffusivity

Authors: This should now be clear to readers as we have added values for diffusion coefficients as above, and the differences in S between DIN and DIP species are a proportion of D.

Ref1: p. 7, L 27: Quantify temperature effects on S (don't need a lot here, but something to give readers a sense if the error from ignoring temperature variability is on the order of 0.01%, 0.1%, 1%, 10%, etc. would be useful).

Authors: Keeping temperature constant rather than allowing it to vary (and change viscosity) would change values of S by 0.009%. We have added this value to the ms (p8 L2).

Figures: Ref1: Fig 1: Do you have example photos from the SG and CR sites that could be added to this figure to help convey the communities described in the 1st paragraph of the Methods? I think this visual representation would help readers understand the two sites.

Authors: We have added two photos from the sampling sites to Figure 1.

Ref1: Fig. 4: Please describe the error bars (e.g., SD, SE, 95% CI, etc.)

Authors: Thanks for picking that up! Those are standard deviation, which has been added to the figure caption.

Ref1: Fig. 5: I think it would be interesting to put dashed lines on these plots to show

the range of S estimates from flume and wave-driven field system studies (p. 10, L 24-28). These would really help show that the variability in S in tidal-dominated systems is far larger than in previously studied systems.

Authors: We have added some lines on Figure 5 and some explanatory text in the figure caption showing a range of S values from previous work.

Ref1: Figs. 5 and 6: I think these two figures should be combined, and it would really help readers to see them as a multi-panel plot so that they can understand how closely the J_MTL estimates mirror the S estimates (or alternatively, depart from each other).

Authors: Earlier versions of this ms had Figs 5 and 6 combined, but we found it confusing as S could get mistaken for a benthic flux estimate since the phase averages look fairly similar. We prefer to leave these figures separate and no changes have been made to the ms.

Tables: Ref1: Table 1: "Number of duplicate samples"- does this mean the total number of replicates analyzed? Or the total number of unknown water samples collected, each of which were duplicated? Please clarify. Tables 1 and 2: I think these tables can be combined. This would streamline the manuscript by reducing the number of tables (as it stands right now, Table 1 adds little unique information).

Authors: We have combined Tables 1 and 2 together and clarified the distinction about duplicates in the table caption.

Grammar/typos: Ref1: p. 1, L 30: Correct subscripting/superscripting of NOx and NH4+

Authors: Thanks for picking that up! Addressed.

Ref1: p. 7, L 24: "through" (though)

Authors: Thanks for picking that up! Addressed.

**BGD**

Please also note the supplement to this comment:
https://www.biogeosciences-discuss.net/bg-2018-413/bg-2018-413-AC1-
supplement.pdf

―――――――――――――――――――

Interactive
comment

**Supplement:**

[revised manuscript text omitted]

---

## Author Comment (AC2) · 4 Apr 2019

MS No.: bg-2018-413, Anonymous Referee #2

Referee comments are included below denoted as "Ref2".

Author response to each referee comment is given below the comment denoted as "Authors". Please note that page/line numbers correspond to the revised version of the manuscript rather than the original version. Please also note that this version of the manuscript contains corrections from Referee #1.

Ref2: This MS reports fluxes of dissolved inorganic nitrogen and phosphorus and theoretical mass-transfer-limited uptake rates on a strongly tide-dominated reef platform. The amount of nutrients that is released in the water column is calculated from these

two data sets.

General evaluation: Overall, this is a very interesting paper, showing that mass-transfer-limited uptake rates may vary by an order of magnitude on the scale of minutes to hours on tide-dominated reefs, due to substantial variability in flow speeds and water depths over the tidal cycle. Differences between wave- and tide-dominated reef biogeochemistry that are due to the hydrodynamic regime are nicely highlighted. I have a number of relatively minor comments aimed at clarifying the methods and results (detailed below), which the authors should be able to answer easily. The main concern is that nutrient concentrations were not measured in Feb during the first 6 hours of the tidal cycle. Missing data were replaced by nutrient concentrations in offshore waters, which is maybe not perfectly supported by the data, particularly with regard to NOX (Figure 3). I would therefore recommend more caution in the conclusions regarding seasonal differences in JMTL (see below). The discussion about the implications in terms of nutrient limitation is worthwhile, however.

Authors: We thank the referee for their supportive and thorough comments on this manuscript (ms). We address the concerns about using offshore waters for nutrient concentrations in specific comments below.

Comments in detail: Introduction

Ref2: p. 2 Line 5: you could add a reference about corals: Grover, R., Maguer, J. F., Alle- mand, D., & Ferrier-Pagès, C. (2008). Uptake of dissolved free amino acids by the scleractinian coral Stylophora pistillata. Journal of Experimental Biology, 211(6), 860-865.

Authors: This reference has been added to p2 L21.

Methods

2.1 Field site

Ref2: p. 4 Lines 8-10: The hydrodynamic study of Lowe at al. (2015) was performed in

March-April 2014, while the MS reports nutrient concentrations from October 2013 and February 2014. Although this is justified later in the MS, the reason why hydrodynamic data collected at the same time as nutrients were not used to calculate mass-transfer-limited uptake rates is unclear at this point.

Authors: We moved an explanatory sentence from p6 to p4 L10-13 to clarify this point earlier in the Methods section for readers.

Ref2: p. 4 Lines 11-13: It is unclear if this concerns mass-transfer-limited uptake rates only, or nutrient fluxes as well.

Authors: We have added clarification to p4 L14 to indicate that this applies to all data.

2.3 Control volume approach

Ref2: Line 8: "Depth-averaged flow speeds were bin-averaged": do you mean that you first averaged flow speeds in each bin (at 5 min intervals) before depth averaging?

Authors: We think the word "bin-averaged" may have been confusing, so we have clarified the wording on p5 L11.

Ref2: Lines 13-21: I think this paragraph needs some clarification. You are using the mean of concentrations at both stations to calculate the local benthic flux, but you explain then that this local term represents nutrient uptake or release occurring at a sampling station (in my mind, CR or SG, while you are calculating Jnet on the transect). The advective flux is described as nutrient uptake or release during transit between sampling stations, which I also find very confusing. Should not it be what is added or subtracted to the transect due to water transport?

Authors: A control volume approach comes from fluid dynamics and is using both Eulerian (the "local" flux term) and Lagrangian (the "advective" flux term) frames of reference. Basically, when you take samples at fixed positions in a moving fluid, the changes that you see are always a balance of changes in situ (i.e., nutrient uptake at your sampling station) and changes due to water masses advecting into your sampling

station. Depending on the flow speed, one term may dominate over another (e.g., in fast flow, the "advective" term will dominate, while in slow flow the "local" term will dominate). Control volumes are used to estimate net fluxes over the entire volume (so a mix of coral and seagrass on the reef flat). We have clarified this on p5 L22-23.

Ref2: The minus sign in front of Jnet is also surprising at first glance. It might be more understandable to state that the sign of Jnet was reversed so that uptake is positive and release negative.

Authors: The minus sign is typical of benthic flux studies, as it is used to define the frame of reference. There is an explanation of the sign convention on p5 L21-22.

2.4 Uptake rates at the limits of mass-transfer

Ref2: p.5 Lines 25-30: In the results, JMTL is first calculated for both CR and SG (Figure 6), not along the study transect. Maybe the reasoning would be easier to follow if there was first a paragraph about the calculation of mass-transfer velocity and JMTL at each one of the two stations, and then, a new paragraph explaining the calculation of Jrelease.

Authors: In Section 2.4, JMTL is first calculated along the study transect (p6 L4) similarly to how Jnet integrates fluxes along the study transect. Addition of text to p5 L22-23 should clarify that Jnet is an integrated measurement over the transect, which should alleviate confusion. On p6 L23-26, Jrelease is calculated using Jnet and JMTL. At the end of this section (p6 L28-p7 L6), we calculate JMTL over the full tidal cycle at both stations individually; we have added some text to this section to clarify these calculations.

Ref2: p.5 Line 30 to p.6 Line 9: This, although necessary, is really hard to follow. There are very nice explanations given in the results (p. 7 Lines 23-27) that might help the reader to go through the equations. Would it be possible to integrate these explanations into this paragraph, specifying the parameters whose variability has the

most influence?

Authors: We have moved some text from the Results to p6 L9-12 to give a simpler explanation of mass transfer velocity.

Ref2: Maybe, in a second step, you could simplify equation 4, as some parameters are constants (or nearly constant). This would highlight the influence of flow speed (and possibly water depth: the drag coefficient CD was taken as 0.02 in Gruber et al. (2017), but I didn't understand if this is the case here) and nutrient diffusivity?

Authors: We prefer to leave Eq. 4 as it is; constant variables are defined in the text below each equation, so it should be clear which variables are constant (very few in Section 2.4 – just density, kinematic viscosity, and diffusivity). We have added the constant values for diffusivity to p6 L13 to make this easier for readers to use.

Ref2: In Gruber et al. (2017), u* is a function of ux, not u (this MS). Is there an explanation? I would also suggest that you explain why you are using u in S calculation, and not ux.

Authors: Ux is used when we make estimates (such as with the CoVo) during the ~unidirectional ebb tide period (since ux is flow speed along the axis of the transect). When we calculate JMTL over the full tidal cycle (no longer ~unidirectional, but rather with large changes in direction), we use u (non-rotated flow speed). We clarified this by using ux in Eqn 4,7 and then explaining the difference in text p7 L1-3.

Ref2: Lines 16-28: I would suggest to give these informations before Line 10 (calculation of Jrelease). Could you please clarify which instruments were deployed exactly? On Figure 1, a velocimeter is shown at CR, but you are not using the data, right? Jnet was calculated from the ADCP data at SG, are you using the same current speed data for the calculation of JMTL and then Jrelease, or the data from Lowe et al. (2015)?

Authors: We think this should now be clarified due to clarification of JMTL calculation at the beginning of Section 2.4 (see responses above). ADP data were used to calculate

Jnet, JMTL, and Jrelease as this instrument was deployed in Oct, Feb, and Apr. ADV data were only available from April, so were used only to calculate the full tidal cycle version of JMTL (last part of Section 2.4). We have added explanation of which velocity data were used on p6 L31-p7 L1 and added a comment about ADV deployment time into Figure 1.

Ref2: Lines 29-30: This deserves more explanations. Do you mean, for example, that error bars on Figure 4 are uncertainties in estimates of each one of the calculated fluxes? Please add this information in the caption.

Authors: Indeed, the error bars in Figure 4 are standard deviations of each calculated flux. We have added this information to the caption. Good catch!

Results 3.1 Nutrient concentrations and measured fluxes

Ref2: Line 4: From Table 2, water temperature is about 2âŮeC warmer in Feb (not 3âŮeC). Some observations could be supported by statistics: line 6 (DIP and DON are slightly lower in Feb); lines 6-7 (concentrations are similar on the reef and offshore until 6 hours after flooding); lines 18-19 (no difference between Oct and Feb for NH4?).

Authors: Good observation about temperature, we have corrected this. We don't think statistics on nutrient concentrations or fluxes would add very much to this ms. The purpose of this ms is not to determine whether seasons are significantly different from one another. If that were our purpose, we would need to replicate by season (at least 3 dry seasons and at least 3 wet seasons, which we did not do). Doing statistics would give you some p-values, but it wouldn't actually give you any truth about seasonality.

3.2 Mass-transfer velocity and nutrient uptake

Ref2: It is unclear from p. 6 lines 3-5 that the drag coefficient was calculated as a function of water depth to draw Figure 5 (the drag coefficient was taken as 0.02 in Gruber et al., 2017). Please clarify in the Methods.

Authors: We actually used a variable form of CD (from McDonald et al. 2006), which

was an accidental omission from the ms. We have added an equation and explanation for calculating CD (p6 L14-18). Between the publication of Gruber et al. 2017 and the submission of this ms, we found a more realistic (we believe) way of estimating the drag coefficient. Drag in shallow reef environments is very much an ongoing field of research!

Ref2: p. 7 Lines 26-27: Can you roughly quantify the effect of temperature on S?

Authors: Temperature changes alter S by <0.01%, and we have added this value to p8 L12.

Ref2: p. 7 Line 28: Could you add on Figure 5 the drag coefficient as a function of hours after reef flooding (and maybe flow speed as well, to avoid having to go back to Figure 2). Also state in the caption that hydrodynamic data (and presumably water depth?) are from March-April 2014 (Lowe et al., 2015), while temperature and salinity from Oct 2013 and Feb 2014 (is that right?).

Authors: We have added 4 more panels to Figure 5, which now shows tidal phase-averages of CD and u for both communities. We have also added the clarification that hydrodynamic data are from April 2014 to the caption. Temperature is from April as well, but as discussed above, temperature affects S by a negligible amount (<0.01%); salinity is not part of these calculations, except in the viscosity of seawater, which effectively a constant over the range of salinities measured in the coastal ocean.

Ref2: p. 7 Line 32: Figure 2 doesn't show water depth at SG and CR. Could you add tidal phase-averaged water depth at each site on Figure 3?

Authors: Figure 2 shows water depth at SG, and we have added this clarification to the caption of this figure. Water depth at CR is indistinguishable from SG as the reef platform is basically flat (there is a 10 cm vertical difference between SG and CR, which would not be visible in these figures). Tidal phase-averaged water depth is already shown in Figure 3 as a black line.

Ref2: p. 8 Line 4: Could you explain "which differ by a factor of <4"? Is it the ratio of the flow speeds?

Authors: This statement refers to S, and we have added this clarification to p8 L21.

Ref2: Figure 6: There are missing nutrient data, especially in Feb (0-6 hours after reef flooding; Figure 3). I understand from p. 6 Lines 24-26 that missing data were replaced by nutrient concentrations in offshore waters. From Figure 3, this looks acceptable in Oct, but maybe less in Feb, especially for NOx. Could you show on Figure 6 the time periods during which nutrients were actually measured?

Authors: As the referee points out, we did not collect water samples during flood/high tide in Feb. However, from the samples collected in Oct, it can be seen that water flooding the reef (i.e. at 0-1 hours after reef flooding in Figure 3) has nutrient concentrations very close to offshore waters. This is because this flooding water is offshore water – flow is occurring across the entire reef flat, which is why seagrass and coral nutrient concentrations are similar during this time. As can be seen in Oct, sometime around high tide (∼3 h after reef flooding) concentrations begin slowly diverging from those in offshore waters. This same pattern of reef concentrations roughly matching offshore concentrations from 0-3 h would almost certainly occur in Feb as well, since the hydrodynamics of the reef remain the same. We agree that there is a period of 2.5 hours (3 – 5.5 hours after reef flooding) when concentrations were not measured but are likely diverging from those offshore. Rather than make assumptions about what nutrient concentrations were during those 2.5 hours, we applied rather large error terms to tidal phase-averaged concentrations of NOx and DIP (standard deviations of 0.5 uM and 0.05 uM, respectively). These values are given on p7 L17-18. This error was then propagated (via Monte-Carlo) through all estimates of JMTL.

Ref2: p. 8 Line 10: JMTL is not shown for ammonia. Why?

Authors: JMTL isn't shown in the large multi-panel figures (Figs 4, 5, and 6) because it looks very similar to NOx (being concentration and flow-dependent) and we were trying

to minimize the size and complexity of figures (there are quite a few figures already). The reference to NH4 on p8 L27 has been removed to clarify this sentence.

Ref2: p. 8 Lines 11-13: This looks speculative, as nutrient concentrations were not measured in Feb during the first 6 hours of the tidal cycle (see previous comment on Figure 6). Please state clearly that you are assuming that NOx concentrations are similar on the reef and offshore during this period and add some comment in the Discussion.

Authors: We are confident that for the period from flood to high tide (0-3 h after reef flooding in Fig 6), offshore nutrient concentrations are representative of concentrations on the reef for reasons discussed above. We have made the wording more cautious on p8 L28-30.

Ref2: Figure 4: I assume that JMTL is the mean of the values shown on Figure 6 for SG and CR?

Authors: JMTL shown there was calculated along the transect (SG to CR). We have added a note in Figure 4 caption to clarify.

Ref2: p. 8 Line 17: Is "NOx release" "net NOx release"? Seasonal differences: could you add your stats as a column in Table 3?

Authors: We have added "net" to NOx release as suggested.

p. 8 Line 19: This comment refers to Lines 11-13 (see above). I would suggest to simply state that, contrary to DIP, DIN concentrations are similar at both seasons during the part of the tidal cycle studied.

Ref2: P 8 from Line 21, and Figures 7 and 8: I understand that you averaged S and JMTL over each one of the 12-hours period available, and then averaged these averages. If that's right, first you should explain why, and then, I don't think averaging averages is the best way to assess standard deviations (they also appear very small in Figure 7, given the range in Figure 5).

Authors: That is correct. We thought this would be an interesting way to conceptualise the data in a 'bigger picture' sense given that tide-dominated systems are so physically controlled by the tide. We could have just averaged all the data together, but we were trying to be creative (and more physically-focused) since tide-dominated systems have never really been studied before! We have added explanation to p7 L8-9. Standard deviations are not averaged, but are generated by running the entire calculation (as shown in the Methods) with 10,000 sets of noise-corrupted data using Monte-Carlo simulations. The range you see in Figure 5 is not standard deviations, but are values from the spring vs neap cycle (see Fig 5 caption). Standard deviations are relatively small for our estimates because S is so closely controlled by flow speed (as you can see in Fig 5); as a result, JMTL is also closely controlled by flow speed. Flow speed has a small standard deviation thanks to the accuracy and precision of ADPs/ADVs, which is then propagated through our estimates.

Ref2: Lines 21-24 and Figure 7: I'm not sure this is very useful. The two points about S are: (1) the small difference between SG and CR, which is already described p. 7 Lines 31-32 and p. 8 Line 3, and (2) the difference between DIN and DIP which you can easily talk about after p. 7 Lines 25-26.

Authors: Figure 7 does add to the story because it is showing the 'bigger picture' of mass transfer velocity. The first discussion of S (p7 L31-32 in the referree's comment) talks about how it varies within a tidal cycle (at times 30% greater at CR vs SG). Figure 7 (and Lines 21-24 in the referee's comment) show the 'bigger picture' that when averaged over longer time periods, differences between SG and CR are actually fairly small. We prefer to leave this as is.

Ref2: From Line 24 and Figure 8: Again, I would not recommend using all data from Figure 6 due to missing nutrient values. Differences between seasons and sites are already described (p. 8 Lines 6-14).

Authors: We believe we are representing missing nutrient values in a reasonable way

through: assumptions based on hydrodynamics (described previously) and relatively large error terms (standard deviation of 0.5 uM for NOx), which are propagated through these calculations. We would prefer to leave this figure as is.

Discussion

Ref2: p. 9 Line 17 (DIP and DON): DIP and DON are "slightly lower (. . .) in Feb" p. 7 Line 6 and "similar between seasons" p. 9 Line 17. One of these two sentences needs to be re-written after stats are performed.

Authors: We have changed p7 L24 to read "similar between seasons". Our response to stats is discussed previously.

Ref2: Line 28 to p.10 Line 2: Do you mean that mass-transfer-limited uptake was demonstrated in controlled environments because nutrient release was negligible compared to uptake?

Authors: That is the most likely explanation why flume experiments show uptake near mass-transfer limits. Whereas, in natural reef environments, a host of other processes (detrital remineralisation, phytoplankton grazing, etc) are occurring that confound uptake measurements (as discussed in the following paragraph in Section 4.2). We have added text (p10 L12) to clarify this.

Ref2: p. 10 Line 4: overestimation of DIN release on Tallon: I don't understand your point. Whatever the source, I think that your calculation of DIN release is fine. Could you clarify?

Authors: This line is simply addressing the question of other DIN inputs to the CoVo that may not be accounted for in our calculations (like N2-fixation and porewater advection, on which there is some existing literature). We think this text is fine as written.

Ref2: References There are two papers by Lowe et al., 2015. Please use 2015a and 2015b throughout the text.

Authors: There is only one Lowe et al., 2015 paper. The other paper is Lowe and Falter 2015.

Please also note the supplement to this comment:
https://www.biogeosciences-discuss.net/bg-2018-413/bg-2018-413-AC2-supplement.pdf

─────────────────────────

**Supplement:**

[revised manuscript text omitted]

---

## Author Comment (AC3) · 4 Apr 2019

MS No.: bg-2018-413, Anonymous Referee #3

Referee comments are included below denoted as "Ref3".

Author response to each referee comment is given below the comment denoted as "Authors". Please note that page/line numbers correspond to the revised version of the manuscript rather than the original version. Please also note that this version of the manuscript contains corrections from Referees #1 and 2.

Ref3: The study reports results from measurements of dissolved nitrogen and phosphorus fluxes in a tide-dominated reef. The fluxes were estimated using a one-dimensional control volume method, i.e. nutrient concentrations were measured on

the reef platform at two points along the flow path of the tidal currents, and from the concentrations changes occurring between the two points, fluxes were estimated. The measurements suggest a release of nitrate, while fluxes of NH4+ and DIP varied between net uptake and release. The results are interesting and produce new insights into the functioning of tide- dominated reefs.

Authors: We thank the referee for their supportive comments and review of this manuscript (ms).

Ref3: A few points need clarification and revision: This is a subtropical environment with healthy seagrass and coral cover. Nutrients in the water column are low and it is surprising that the reef platform with seagrass and healthy coral releases nitrate during low tide. Assuming that during the low water phase photosynthesis of vascular and unicellular plants reaches a maximum, one would expect nutrient uptake of the benthic community resulting in a net uptake.

Authors: Actually, it is common for in situ reef studies communities to measure a release of nitrate. There are many references cited in the Discussion that have also found nitrate release on reefs (p10 L12-16). This is thought to be due to processes such as grazing of benthic filter-feeders and other reef organisms on phytoplankton and detrital remineralisation. To our knowledge, no one has yet conclusively showed why nitrate release occurs, so this is very much an open question! We would guess that this release is not coming from the seagrass community; based on nutrient concentrations at the end of ebb tide (when advection was negligible) shown in Figure 3, nitrate is quite low in the seagrass community. It is important to note that the control volume (CoVo) method provides estimates of benthic fluxes over the length of the transect (thus a function of all ecological communities along the transect), rather than separate fluxes for seagrass and coral communities. This point has been clarified on p5 L22-23.

Ref3: Tidal current vector fields previously published by the authors for this study site (Lowe et al. 2015) revealed that the tidal water in and outflow is not symmetrical,

**BGD**

Interactive
comment

resulting in a residual current entering the reef at its southeastern edge and leaving at its northeastern edge. These residual currents include flows parallel to the reef, which may cause that the two sampling stations measure some waters that took different pathways over the reef platform. It is conceivable that the residual currents transport water with different nutrient concentrations to the two measuring stations and that the observed difference in nitrate concentration between the two stations is a function of the pathways of the residual currents.

Authors: We are confident that 'current pathways' are not the cause of the observed nitrate release. The CoVo approach is only used when the direction of flow aligns with the sampling transect (during ebb tide), and not when flow vectors are moving perpendicular to the transect (this is described on p5 L6-8). Nitrate concentrations become most elevated at the end of ebb tide, when flow is very slow ∼2 cm/s. During this period, the "local" flux term will dominate our estimates of Jnet, meaning that local processes (nutrient release in the vicinity of the station) are causing these changes rather than advection of water parcels.

Ref3: The reef lagoon appears to be lined by mangrove forest and the seagrass community accumulates organic-rich sediments. The mixed zone between seagrass and coral has pockets of sediment and a porous structure. Could release of fluid from the sediments and the porous structure of the mixed zone explain the nutrient increase during decreasing water level as the path of the residual currents is along the mixed zone? This should be clarified.

Authors: There may be porewater advection occurring on Tallon reef, which we discuss on p10 L16-18. However, we doubt this is the cause of increases in nitrate, as porewater tends to have very low nitrate concentrations (near detection limit). The cause of net nitrate release is most likely the processes discussed in Section 4.2.

Ref3: The tidal range exceeding 8 m is unusual. This is not a typical scenario, which needs to be considered and pointed out when generalizing the results.

Authors: We agree with this comment and have added a new paragraph to the Discussion to help readers understand how our results can perhaps be generalised to other reefs that are tide-dominated but not necessarily macrotidal. See p12 L19-28.

Minor comments.

Ref3: P1L9 a "forcing" is not a "regime", please rephrase.

Authors: We have rephrased this statement.

Ref3: P1L25 "Reef waters have carbon concentrations that are orders of magnitude greater than nitrogen (N) and phosphorus (P)". If a ratio close to Redfield is applicable here, C concentrations order(s) magnitudes higher than those of the nutrients can be expected. Benthic communities would not be nutrient limited. If carbon is way higher, please rephrase.

Authors: I'm not sure I understand this comment. P1 L25 is referring to the fact that benthic organisms tend to be N and/or P limited rather than C-limited. The Redfield ratio is a C:N:P ratio that is often used to infer phytoplankton nutrient limitation, and I'm not sure how this is related to the line in our ms?

Ref3: P2L10 "labile dissolved inorganic species". "Labile" here seems the wrong word as some of these inorganic species can be very stable in the marine environment.

Authors: "Labile" is a term commonly used to refer to NOx, NH4, and PO4, as these nutrient species can be readily utilised by organisms. In contrast, DON and DOP are often called "refractory" meaning less-readily utilised by organisms.

Ref3: P2L16 "turbulent transport" should be added to the list of the controls of nutrient transfer

Authors: This sentence is referring to the relationship derived from fluid dynamics that controls benthic nutrient uptake and is not a 'list of controls of nutrient transfer'. Later in the ms (Eqns 3 and 4), these relationships are defined mathematically, and do not

include turbulence.

P4L8 "reef benthos, which represent the net uptake or release of nutrients". Please add information on nutrient uptake/release of water column organisms

Authors: This statement has been rephrased (p4 L7-8) for clarity.

Ref3: P4L32 "All nutrient concentrations presented are the mean of duplicate samples" why didn't you take triplicate samples, which would have allowed calculation of standard deviation, opening up other options for statistical analysis?

Authors: Triplicate samples weren't part of this study design, as we weren't attempting to determine statistically whether one nutrient measurement was significantly different to another nutrient measurement. Error estimates, however, are central to this study as described on p7 L14-18.

Ref3: P5L8 How large is the error introduced by using depth-averaged current velocity instead

Authors: Instead of...? Depth-averaged current velocity is fairly ubiquitous in studies of shallow water environments.

Ref3: P11L8 If the coral zone is 20% more productive than the seagrass zone (Gruber et al., 2017), one would expect an increasing N consumption during decreasing water level as light intensity at the reef surface increases, with higher N demands in the coral zone. The results suggest the opposite, how is this explained?

Authors: As mentioned above, the CoVo approach gives flux estimates over the full transect (incorporating seagrass and coral communities). This release of nitrate is most likely coming from the reef community, for reasons that are discussed above (phytoplankton grazing and remineralisation) and in detail in Section 4.3 of the Discussion.

Ref3: P11L20 "In a simplified wave-driven reef, offshore (oceanic) water moves from

reef crest to back reef roughly unidirectionally. Thus, benthic communities are subjected to the physico-chemical water properties present in offshore waters modified by the communities 'upstream' of them." This should be explained in more detail, as water transported into the reef also has to leave the reef, irrespective of the transport process. This release may traverse the communities that contacted this water before as in the tidal dominated reef.

Authors: On wave-driven reefs, water tends to cross the reef flat and then travel alongshore and eventually exit the reef through channels. This is a well-established transport process (see Monismith, S. (2007): Hydrodynamics of coral reefs, Annu Rev Fluid Mech, 39, 37-55 for more details and references therein). We have added some text (p12 L5) to clarify this.

Please also note the supplement to this comment:
https://www.biogeosciences-discuss.net/bg-2018-413/bg-2018-413-AC3-supplement.pdf

**Supplement:**

[revised manuscript text omitted]